# Adolescent BCG revaccination induces a phenotypic shift in CD4+ T cell responses to *Mycobacterium tuberculosis*

One B. Dintwe [1,2,6], Lamar Ballweber Fleming [1,6], Valentin Voillet [1,2,6], John McNevin [1], Aaron Seese [1], Anneta Naidoo[2], Saleha Omarjee[2], Linda-Gail Bekker [3], James G. Kublin[1], Stephen C. De Rosa[1,4], Evan W. Newell [1], Andrew Fiore-Gartland [1,7], Erica Andersen-Nissen [1,2,7] ✉ & M. Juliana McElrath[1,5,7] ✉

A recent clinical trial demonstrated that Bacille Calmette-Guérin (BCG) revaccination of adolescents reduced the risk of sustained infection with *Mycobacterium tuberculosis* (*M.tb*). In a companion phase 1b trial, HVTN 602/ Aeras A-042, we characterize in-depth the cellular responses to BCG revaccination or to a H4:IC31 vaccine boost to identify T cell subsets that could be responsible for the protection observed. High-dimensional clustering analysis of cells profiled using a 26-color flow cytometric panel show marked increases in five effector memory CD4+ T cell subpopulations (T_EM) after BCG revaccination, two of which are highly polyfunctional. CITE-Seq single-cell analysis shows that the activated subsets include an abundant cluster of Th1 cells with migratory potential. Additionally, a small cluster of Th17 T_EM cells induced by BCG revaccination expresses high levels of CD103; these may represent recirculating tissue-resident memory cells that could provide pulmonary immune protection. Together, these results identify unique populations of CD4+ T cells with potential to be immune correlates of protection conferred by BCG revaccination.

In 2021, an estimated 1.6 million individuals globally died from tuberculosis (TB), a number that has shown a concerning increase over the previous 2 years[1]. TB drug treatment regimens are taxing on the public health system and there is evidence of continual bacterial evolution towards drug-resistant strains[1]. Morbidity due to TB disease takes a severe toll on individuals in high-burden countries, even after successful treatment[1]. Despite the roll out of anti-retroviral therapy in many countries, people living with HIV are still disproportionately affected by TB, with the highest morbidity and mortality rates in this group[2]. An efficacious TB vaccine is urgently needed to prevent these high levels of morbidity and mortality worldwide.

The only licensed TB vaccine, Bacille Calmette-Guérin (BCG), protects against disseminated disease and death when administered to infants at birth but provides variable protection against disease in adults[3]. Tuberculosis risk changes with age: in school-aged children there is a lower risk of infection as compared to that in infants and young children; the risk increases again in adolescence[4], suggesting this period as ideal for additional, optimized vaccine strategies.

[1]Vaccine and Infectious Disease Division, Fred Hutchinson Cancer Center, Seattle, WA, USA. [2]Cape Town HVTN Immunology Laboratory, Hutchinson Centre Research Institute of South Africa, Cape Town, South Africa. [3]The Desmond Tutu HIV Centre, Institute of Infectious Diseases and Molecular Medicine, University of Cape Town, Cape Town, South Africa. [4]Department of Laboratory Medicine and Pathology, University of Washington School of Medicine, Seattle, WA, USA. [5]Department of Medicine, University of Washington School of Medicine, Seattle, WA, USA. [6]These authors contributed equally: One B. Dintwe, Lamar Ballweber Fleming, Valentin Voillet. [7]These authors jointly supervised this work: Andrew Fiore-Gartland, Erica Andersen-Nissen, M. Juliana McElrath. ✉e-mail: eanderse@hcrisa.org.za; jmcelrat@fredhutch.org

South African adolescents are at particularly high risk of TB infection with a force of infection estimated at 14% per annum in some areas[5]. The Aeras C-040-404 phase 2 clinical trial evaluated the efficacy of two immunization booster regimens for prevention of TB infection in healthy South African adolescents who received BCG vaccine at birth; revaccination with one dose of intradermal BCG, or intramuscular vaccination with two doses of H4:IC31, an Ag85B/TB10.4 fusion protein adjuvanted with an immunostimulatory antibacterial peptide and a synthetic TLR9 agonist[6]. Neither vaccine prevented initial infection (measured by initial QuantiFERON [QFT] conversion). However, BCG revaccination reduced the rate of sustained infection (measured by sustained QFT conversion, indicated by a positive QFT test result 6 months after the initial conversion) with an efficacy of 45.4% ($P = 0.03$)[6]. H4:IC31 vaccination showed a similar trend that did not reach significance (efficacy of 30.5%, $P = 0.29$).

CD4[+] T cells play a critical role in the control of *Mycobacterium tuberculosis* (*M.tb*) infection[7], and vaccination with BCG induces antigen-reactive CD4[+] T cells in non-human primates[8] and in humans[9,10]. Recent efforts to describe *M.tb*-reactive CD4[+] T cells have focused on Th1 and Th17-type cells[11], as well as a subpopulation of CXCR3[+]CCR6[+] helper T cells (termed Th1* cells) that are more abundant among TB-sensitized individuals[12] and increase after BCG vaccination of BCG-naïve adults[13]. However, much remains to be learned about these phenotypic and functional subsets of T cells that are induced or boosted by BCG. Describing the surface markers, functional responses and gene expression profiles that distinguish subsets of T cells modulated by BCG is a critical step towards elucidating their potential role in protection.

In this study, we used high dimensional flow cytometry coupled with CITE-Seq (Cellular Indexing of Transcriptomes and Epitopes by Sequencing)[14] to extensively probe T cell responses in adolescents enrolled in the HVTN 602/Aeras A-042 trial[10], a phase 1b companion study to the Aeras C-040-404 trial. We hypothesized that BCG revaccination boosts memory or induces T cell populations critical for *M.tb* control, and that in-depth analyses of cellular responses could identify the biomarkers associated with these beneficial responses. We show that a subset of naïve-like CD4[+] T cells decreased after revaccination, suggesting that these cells are trafficking to tissues to carry out effector functions or differentiating into the effector cell subsets. BCG revaccination also increased the proportion of circulating polyfunctional *M.tb*-reactive effector memory CD4[+] T cells. Two distinct clusters of effector memory CD4[+] T cells increased after BCG revaccination – one of these may represent a type of recirculating tissue-resident memory T cell that has been shown to contribute to protection against TB in the lung[15,16]. These results provide further insight into the T cell response to BCG revaccination and help guide ongoing efforts to identify immunological correlates of protection (CoP) induced by BCG revaccination.

## Results

Eighty-four QuantiFERON-negative South African adolescent volunteers (54% female) were enrolled in HVTN 602/Aeras A-042. Participants received either a single BCG re-vaccination at day 0 ($n = 24$), two doses of H4:IC31 at days 0 and 56 ($n = 24$), or placebo at days 0 and 56 ($n = 12$)[10].

### BCG revaccination boosts CD4[+] T cell responses

T cell responses to vaccine matched antigens and *M.tb* lysate were assessed via a 26-color intracellular cytokine staining (ICS) assay in previously cryopreserved PBMC from baseline (pre-vaccination on the day of BCG, first dose of the H4:IC31 or placebo administration) and on days 14/28, 70 and 168 post-vaccination, and gated as in Supplementary Fig. 1. Consistent with our previous analysis of data obtained from the HVTN 602/Aeras A-042 trial[10] and results from the Aeras C-040-404 BCG revaccination trial[6,9,10], BCG-reactive CD4[+] T cells – defined as

cells expressing at least 1 of the 7 functional markers measured (IL-2, IFN-γ, TNF, IL-17a, IL-4/IL-13, IL-22 and CD154) following stimulation with BCG – were present at baseline and were significantly increased at 28 days and remained significantly higher than baseline at 70 and 168 days after BCG revaccination (FDR $q < 0.05$, Fig. 1a and Supplementary Data 1). In contrast, BCG-reactive CD4[+] T cell responses were not significantly increased in adolescents vaccinated with H4:IC31 or those who received placebo. No significant differences were observed in the proportion of BCG-reactive CD4[+] T cells between vaccine groups at any timepoint (Supplementary Data 2).

Only a few individuals exhibited CD4[+] T cell responses to the vaccine-matched Ag85B and TB10.4 peptide pools at baseline, consistent with previous reports for the H4 subunit vaccine[10,17,18]. Ag85B and TB-10.4-reactive CD4[+] T cell responses emerged only after the second dose of H4 vaccine (Fig. 1a). Ag85B-specific CD4[+] T cell frequencies were low in individuals from the group revaccinated with BCG but increased significantly at day 28 (FDR $q < 0.05$), consistent with reports that BCG can express Ag85B[19]. The H4:IC31 group had significantly higher Ag85B-specific and TB10.4-specific CD4[+] T cell responses than the BCG revaccination and placebo groups at day 70, as well as Ag85B-specific CD4[+] T cell responses at day 168 (FDR $q < 0.05$, Supplementary Data 2).

To more broadly characterize prior sensitization to BCG, *M.tb* and/or non-tuberculous environmental mycobacterial species, we examined responses to *M.tb* lysate. As expected, *M.tb* lysate-reactive CD4[+] T cells were abundant at baseline in all groups, showing a median response ≥0.1%. No increases in the frequency of *M.tb* lysate-reactive CD4 + T cells were observed in the placebo group (Fig. 1a). The frequency of *M.tb* lysate-reactive CD4[+] T cells significantly increased after both BCG revaccination at days 28, 70 and 168, and H4:IC31 vaccination at day 70 (FDR $q < 0.05$); BCG revaccination induced a durable response with the frequency of *M.tb* lysate-reactive CD4[+] T cells remaining above baseline levels at day 168 (FDR $q < 0.05$). At both days 14/28 and 70, the BCG revaccination group had significantly higher *M.tb* lysate-reactive CD4 + T cell responses than the H4:IC31 group, but not the placebo group (FDR $q < 0.05$, Supplementary Data 2).

Because CD8[+] T cells exhibited limited responses to the antigens tested and no significant changes in frequency occurred post-vaccination (Supplementary Fig. 2), our results focus on CD4[+] T cell responses.

### Vaccination boosts polyfunctional CD4[+] T_EM cells

CD4[+] T cells are critical for protection against TB[20–22]. A durable and protective TB vaccine in humans is thought to require the induction of polyfunctional effector memory CD4[+] T cell populations that respond rapidly in the lung[23] and/or long-lived CD4[+] T cells that exhibit a less-differentiated central memory phenotype[24–28]. Our 26-color flow cytometry panel allowed us to determine the effect of vaccination on these memory T cell subsets.

The effector memory subset (CD45RA[−]CCR7[−], T_EM) of BCG-reactive CD4[+] T cells predominated at baseline, and the same subset increased significantly in frequency after revaccination (Fig. 1b and Supplementaryl Data 1). The frequency of BCG-reactive central memory (CD45RA[−]CCR7[+], T_CM) CD4[+] T cells was very low but also significantly increased after BCG revaccination (FDR $q < 0.05$, Fig. 1b and Supplementary Data 1). Vaccination with two doses of H4:IC31 induced and significantly increased CD4[+] T_EM, T_CM and CD45RA[+]CCR7[−] terminally differentiated effector memory (T_EMRA) responses to both Ag85B and TB10.4 (Fig. 1b), with T_EM being the dominant population. A significant increase in *M.tb* lysate-reactive T_EM cells was observed at day 70 in the H4:IC31 group (Fig. 1b and Supplementary Data 1).

*M.tb* lysate-reactive CD4[+] T cells at baseline were predominantly T_EM and their frequency also increased after BCG revaccination (Fig. 1b). The three other subsets of *M.tb* lysate-reactive CD4[+] memory T cells were also present at baseline; T_CM and T_EMRA cells, but not naïve-

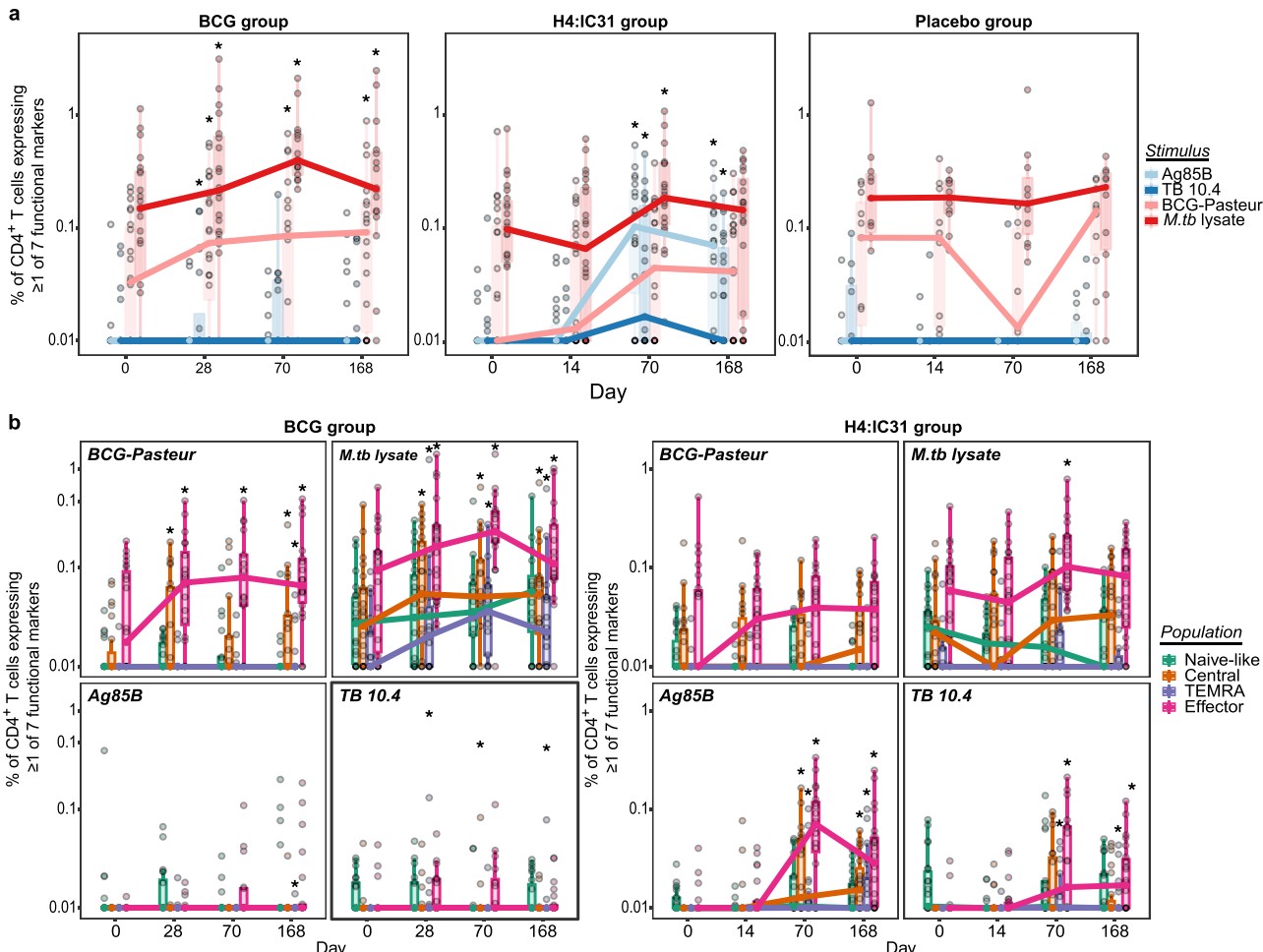

**Fig. 1 | BCG revaccination and H4:IC31 vaccination boost effector memory CD4⁺ T cell responses in South African adolescents.** Cellular responses were evaluated by 26-color intracellular cytokine staining (ICS). **a** Proportion of antigen reactive CD4⁺ T cells in the BCG revaccinated ($n = 22$ biologically independent samples), H4:IC31 vaccinated ($n = 24$ biologically independent samples) and placebo groups ($n = 10$ biologically independent samples) over time. Circles represent an individual participant's response. Percentage of CD4⁺ T cells expressing at least 1 of the 7 functional markers (IL-2, IFN-γ, TNF, IL-17a, IL-4/13, IL-22 and CD154) is shown after stimulation with the Ag85B (light blue) or TB10.4 (dark blue) peptide pools, or BCG (light red), or *M.tb* lysate (dark red) in the ICS assay. The exact *p* values can be found in (Supplementary Data 1b). **b** Proportion of antigen-reactive memory CD4⁺ T cell populations (naïve-like [CD45RA⁺CCR7⁺], central memory [T$_{CM}$; CD45RA-CCR7⁺], effector memory [T$_{EM}$; CD45RA⁻CCR7⁻], terminally differentiated effector memory [T$_{EMRA}$; CD45RA⁺CCR7⁻]) of total CD4⁺ T cells after BCG revaccination ($n = 22$ biologically independent samples) or H4:IC31 vaccination ($n = 24$ biologically independent samples) following stimulation with the antigens noted in the legend. Paired analysis between timepoints was performed using the two sided Wilcoxon signed-rank test, *FDR q-value $p < 0.05$. Boxplots indicate the median response and the first and third quartiles; whiskers extend to no further than 1.5 times the interquartile range. Source data are provided as a Source Data file.

like cells (CD45RA + CCR7 + ), increased post-BCG revaccination (Fig. 1b and Supplementary Data 1). These data, like those from the C-040-404 trial[9], confirm that BCG revaccination of adolescents boosts existing memory CD4⁺ T cell populations and likely primes new cell populations, some of which may be important for protection[6].

To identify unique subsets of antigen-reactive CD4⁺ T cells responding to vaccination, we conducted an unsupervised analysis of the ICS data using all 20 T cell markers in our ICS staining panel. Network-based clustering[29] of the 297,859 CD4⁺ T cells that expressed at least 1 of 7 cytokines revealed 29 distinct clusters of cells; cells were subsequently merged into 20 clusters on the basis of combinatorial binary expression of markers in the panel (e.g., IL-2+ vs. IL-2-; see Methods for details). We constructed a heatmap displaying the percentage of cells positive for each marker in each cluster, based on our manual gating of the flow cytometry data; the median number of cells per participant from each group is shown by cluster (Fig. 2a). To determine how revaccination with BCG or vaccination with H4:IC31 altered subpopulations of antigen-reactive CD4⁺ T cells, we then

examined how the size of each T cell cluster changed after vaccination. Five clusters of CD4⁺ T cells showed significant increases after BCG and/or H4:IC31 vaccination (B, C, D, F and I) and are summarized in a heatmap (Fig. 2b and Supplementary Data 3). Longitudinal boxplots show the cluster frequencies for each individual to the respective vaccine-matched antigens for these five significant clusters (Fig. 2c and Supplementary Data 3).

All of the clusters induced after vaccination consisted of a majority of T$_{EM}$ cells, however T$_{CM}$ cells were also present in these clusters, suggesting similarities in phenotype of the two memory subsets. Cluster C was highly polyfunctional with almost all cells expressing IL-2, IFN-γ, TNF and CD154 (Fig. 2a). 41% (95% CI 40.35–41.37) of cells in Cluster C also co-expressed CCR6 and CXCR3, a population that has been defined by other groups as Th1* cells and have been implicated in control of *M.tb*[12]. Cells in Cluster B were also polyfunctional but did not express IL-2; in addition only 57% (95% CI 56.51 – 58.36) expressed CD154. Cluster D only increased at day 70 post-vaccination in response to vaccine-matched antigens (Ag85B in

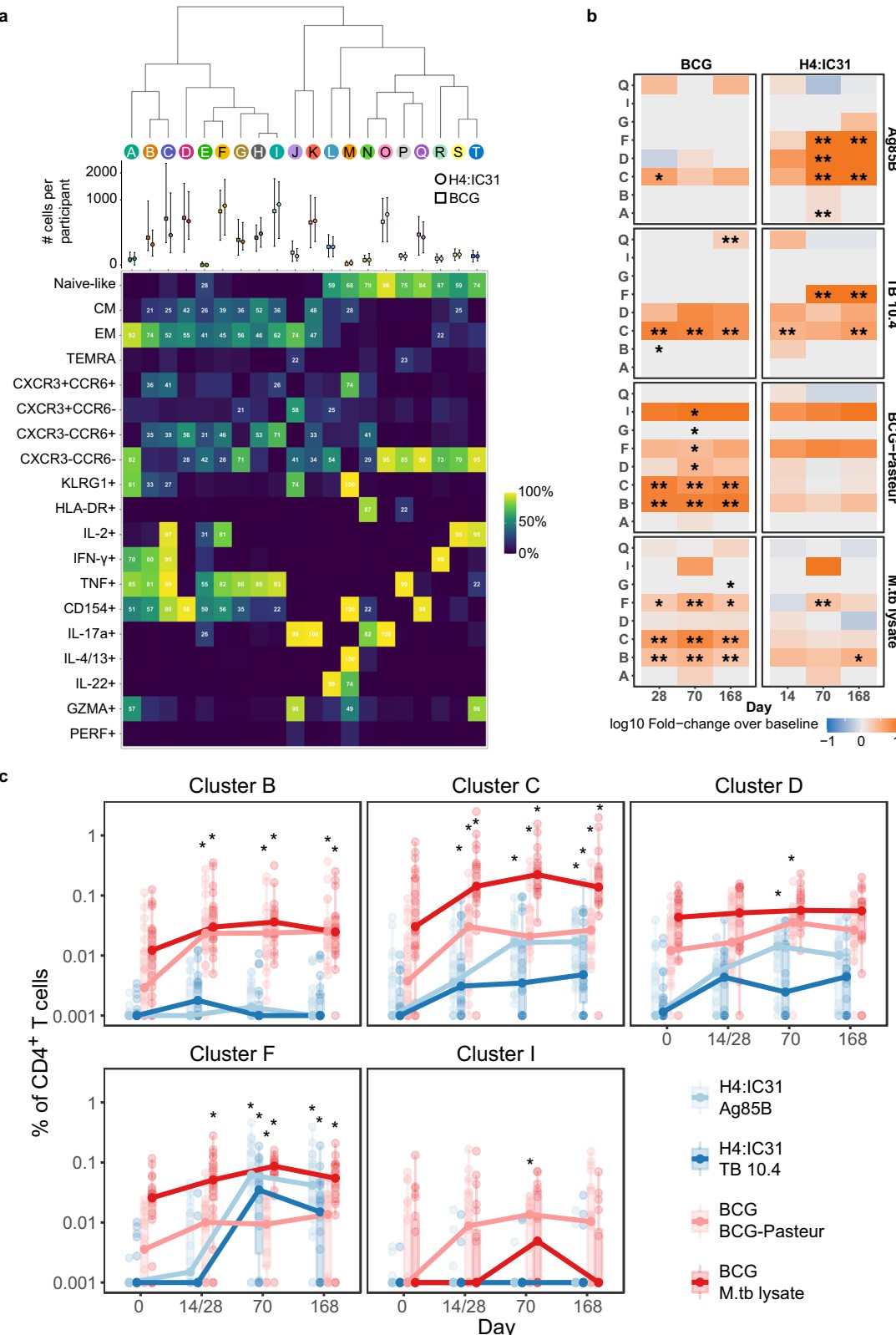

the H4 group; BCG in the BCG group), these cells expressed CD154 without cytokines and 56% (95% CI 55.39 - 56.5) expressed CCR6, a marker highly expressed on Th17 cells[30]. Cells in Cluster F were highly induced after the second dose of H4 vaccine and expressed IL-2 and TNF, but no IFN-γ; 46% (95% CI 45.66 – 46.57) of cells in this cluster expressed CCR6. Cluster I was of low frequency and was only induced on day 70 in the BCG group.

**Antigen-reactive T cells primarily cluster into two subsets**

To further investigate the phenotypes of vaccine-induced T cells, we conducted single-cell analysis of activated T cells using the CITE-Seq method[14]; this approach enables interrogation of many more single cell characteristics than the ICS assay because it couples cell surface protein marker identification using oligo-tagged antibodies – Ab-derived tags, or ADTs – with transcriptional profiling. The ICS and

**Fig. 2 | Multiple clusters of polyfunctional effector memory CD4⁺ T cells increase after vaccination. a** Based on flow cytometry data, heatmap showing the proportion of cells within the 20 PhenoGraph clusters expressing each marker. Each column represents a unique cluster in one or both vaccine groups. The percentage of cells in a given cluster expressing each of the markers evaluated is shown numerically; markers expressed in fewer than 20% of the cells are not annotated. Graph above the heatmap shows the median number of cells per participant in each cluster by vaccine arm with error bars extending to no further than 1.5 times the inter-quartile range. **b** Heatmap showing fold-change in the frequency of CD4⁺ T cells in each of the clusters that showed a significant change over pre-vaccination (Day 0) in at least one timepoint in at least one of the vaccine groups. *FDR $q < 0.05$; **FDR $q < 0.01$. **c** Percentage of total activated CD4⁺ T cells for each significant cluster over the time course. Ag85B (light blue) and TB10.4 (dark blue) peptide pool stimulations; for the BCG group ($n = 22$ biologically independent samples), only the BCG (light red), *M.tb* lysate (dark red) stimulations. *FDR $q < 0.05$, two sided Wilcoxon Signed Rank Test. Boxplots indicate the median response and the first and third quartiles; whiskers extend to no further than 1.5 times the inter-quartile range. Source data are provided as a Source Data file.

CITE-Seq experimental designs are detailed in Supplementary Fig. 3. PBMC from 17 BCG vaccine recipients were stimulated for 16 h with *M.tb* lysate or PBS and activated T cells were enriched via fluorescence-activated cell sorting using two combinations of activation-induced markers (AIM)[31,32]. PBMC from eight H4:IC31 recipients were stimulated with the Ag85B/TB10.4 combined peptide pool or DMSO (the peptide diluent, as negative control) and sorted as above (Supplementary Fig. 3b, c).

The data were first subjected to multiple quality control filters to exclude dead cells and retain cells with high-quality RNA and expected levels of ADTs (see Methods and workflow depicted in Supplementary Fig. 4). Subsequently, activated T cells were identified based on the expression of genes and proteins in individual CD3⁺ cells that responded to either *M.tb* lysate (187,977 cells) or the H4 peptide pool (53,451 cells). A Weighted-Nearest Neighbors (WNN) approach[33,34] facilitated the integration of gene and protein expression data for clustering, annotation and visualization using UMAP (Supplementary Fig. 5a, b).

The vast majority of activated T cells were CD4⁺ and distributed among two different clusters (Supplementary Fig. 5c). CD3⁺ cells not expressing the AIM markers – i.e., "non antigen-reactive T cells" – resided in distinct CD4⁺ and CD8⁺ T cell clusters (Supplementary Fig. 5c). As with the ICS assay, only a small fraction of the activated cells exhibited characteristics of CD8⁺ T cells, γδ T cells or MAIT cells (Supplementary Fig. 5d); therefore the results below largely focus on the CD4⁺ T cell responses.

To assess broad transcriptional changes associated with vaccination, we conducted a single-cell gene expression analysis to detect differentially expressed genes among the activated CD4⁺CD69⁺CD154⁺ T cells. H4:IC31 vaccination significantly increased expression of 6 genes (*LTB, LTA, S100A11, ANZA2, CISH, TNFRSF4*) and decreased expression of 3 genes (*MT2A, TCF7, CCR7*), a pattern consistent with T cell activation and differentiation (FC > 1.5; FDR $q < 0.01$, Supplementary Data 4). No genes exhibited significant changes after BCG revaccination using these criteria. Trends were observed in both vaccine groups for the upregulation of cytokine and cytotoxic gene transcripts observed by protein in the ICS experiments (Supplementary Fig. 6).

**Antigen-reactive CD4⁺ T cells increase and shift toward T_EM**
We focused our single cell analysis on the ~42,000 CD4⁺CD69⁺CD154⁺ (activated; AIM + ) T cells. WNN clustering of this population identified 11 distinct clusters (Fig. 3a) distinguished by gene (Fig. 3b) and ADT expression (Fig. 3c and Supplementary Data 5). Figure 3d shows a heatmap of the top 10 most highly expressed genes that distinguished each cluster and Supplementary Data 6 lists all genes overexpressed in each cluster relative to all other clusters. Based on the gene expression profiles, Clusters 1 and 2 likely represent naïve-like CD4⁺ T cells due to surface protein and gene expression patterns showing expression of CD45RA, CD62L, CD27 and CD31 without expression of CD45RO. Cluster 3 likely represents T_CM cells, Clusters 4−9 represent T_EM cells, Cluster 10 represents regulatory (T_reg) cells, and Cluster 11 includes NKT and γδ T cells.

To begin to understand the effects of vaccination, we overlayed the CD4⁺CD69⁺154⁺ cells by vaccine group and timepoint on the UMAP

and observed that the majority of *M.tb* lysate-activated cells at baseline (BCG vaccine group) were either located in a large naïve-like cluster (Cluster 1) or in Cluster 5, a population of T_EM cells (Supplementary Fig. 7). At day 70, there were still cells in Cluster 1, but most activated CD4⁺ T cells in the BCG group were now located in and around Cluster 5. H4-specific cells from individuals in the H4:IC31 group, by comparison, were concentrated in Cluster 5 at both timepoints (Supplementary Fig. 7), though they increased in frequency after vaccination.

**Differentiation state distinguishes naïve-like CD4⁺ T cells**
The heterogeneity of antigen-reactive CD4⁺ T cells with a naïve-like phenotype is an area of active investigation. These naïve-like cells express markers associated with naïve T cells but can also express effector T cell molecules; multiple subsets of human naïve-like CD4⁺ T cells have been described following infection with *M.tb* and/or vaccination against TB[35,36].

To determine whether vaccination altered the naïve-like subsets identified, we tested for significant changes in the proportion of CD4⁺CD69⁺154⁺ cells in Clusters 1 and 2 between days 0 and 70 in each vaccine group, noting that Cluster 2 contained only 663 cells. BCG revaccination significantly decreased the proportion of CD4⁺CD69⁺154⁺ in Cluster 1 ($p < 0.05$), but not of those in Cluster 2 (Fig. 4a), whereas H4:IC31 vaccination decreased the proportion of CD4⁺CD69⁺154⁺ cells in both naïve-like clusters ($p < 0.05$ in both).

To better understand potential differences in these two naïve-like clusters, we directly compared their gene and surface protein expression (Supplementary Data 7 and 8). In total, 132 genes were differentially expressed between cells in Clusters 1 and 2 (absolute fold-change >1.5, FDR $q < 0.01$; Fig. 4b and Supplementary Data 8). Cluster 1 showed higher expression of genes related to interferon signaling (*IFIT3, IFITIM3, IFI6, CXCL10*) as well as *TNFSF10* which encodes for TRAIL and is upregulated after TCR signaling[37]. Cluster 2 showed higher expression of genes encoding endoplasmic reticulum stress-related proteins (*DDIT3, DNAJB9, DNAJB11, ERLEC1, HILPDA, HSP90B1, HSPA5, HYOU1, KDELR1, SERP1*), suggesting increased protein secretion in response to TCR stimulation and differentiation[38] (Supplementary Data 8).

Both protein and gene expression of CD62L, a marker upregulated in less differentiated cells, was significantly higher in Cluster 1 (fold change of 1.69, $p < 0.0001$) versus Cluster 2 (fold change 2.15, $p < 0.001$; Fig. 4b and Supplementary Data 8). This, combined with the increased expression of *TCF7* in Cluster 1, which encodes a transcription factor (TCF1) that is more highly expressed in naïve than in differentiated cells[39], suggests that cells in Cluster 1 were less differentiated than those in Cluster 2.

**Vaccination increases heterogenous CD4⁺ T_EM subpopulations**
The gradient of memory CD4⁺ T cell phenotypes defined via single cell analysis techniques and described by other groups in the context of viral immunity[40,41] was also present following revaccination with BCG or vaccination with H4:IC31. The activated CD4⁺CD69⁺CD154⁺ memory T cells expressed a heterogeneous set of surface proteins and genes consistent with T_CM (Cluster 3) and T_EM (Clusters 4−9) subsets of cells. For example, by ADTs, Clusters 5, 6 and 9 displayed classical T_EM

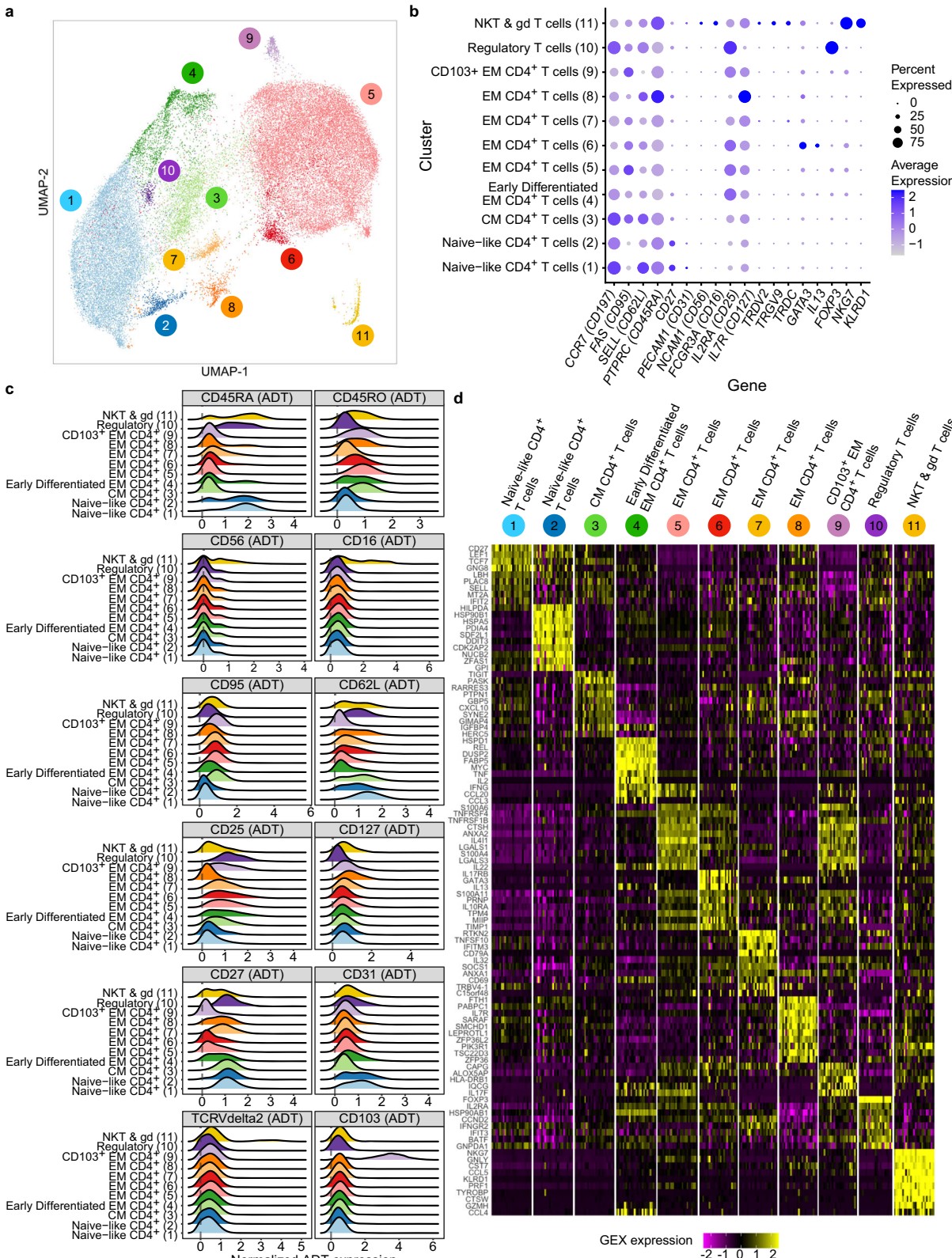

**Fig. 3 | TB vaccination induces distinct clusters of CD4⁺ T cells. a** Unsupervised WNN clustering and UMAP projection of 42,067 CD4⁺CD69⁺CD154⁺ T cells measured by CITE-Seq reveals 11 distinct clusters. **b** Graph showing select genes used in cluster annotation. The percentage of cells expressing select genes (size of circle) and the average expression of each gene (color of circle) by cluster is shown for the CD4⁺CD69⁺CD154⁺ T cells measured by CITE-Seq. **c** Ridge plot of the normalized antibody-derived tagged (ADT) surface marker expression on the CD4⁺ T cells by cluster. **d** Heatmap showing average gene expression per participant of the top 10 genes that distinguish the different clusters from each other.

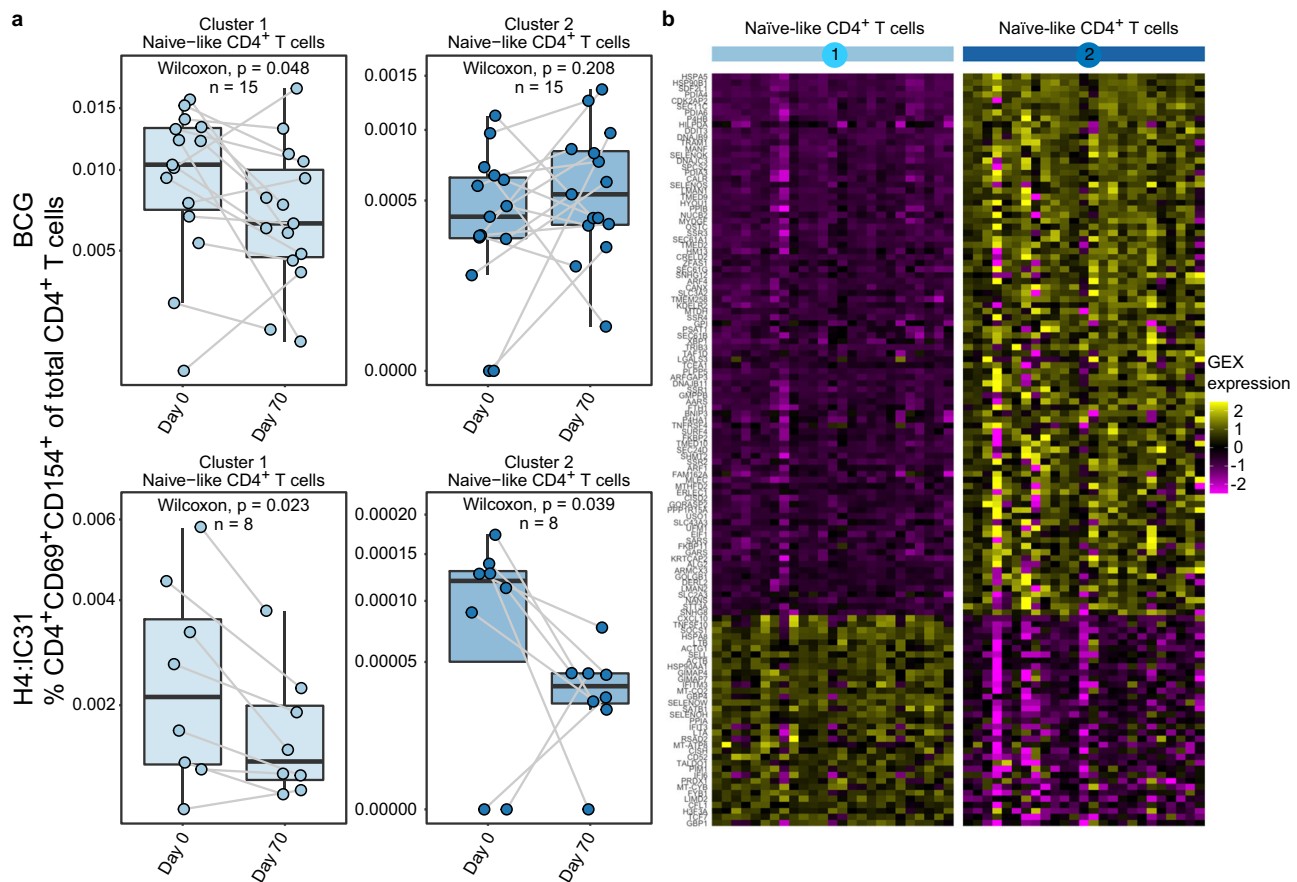

**Fig. 4 | TB vaccination alters antigen-reactive naïve-like cell clusters.**
**a** Proportion of antigen-reactive T cells of total CD4⁺ T cells present in each naïve-like cluster, measured by CITE-Seq, that showed a significant change after vaccination in either the BCG ($n = 15$) or the H4:IC31 ($n = 8$) vaccine groups. *$p < 0.05$, **$p < 0.01$, two sided Wilcoxon Signed Rank test. Each point depicts the response in

a single participant. Boxplots indicate the median response and the first and third quartiles; whiskers extend to no further than 1.5 times the inter-quartile range. **b** Heatmap showing average gene expression by participant (columns) of the genes (rows) that distinguish the 2 antigen-reactive naïve-like CD4⁺ T cell clusters from each other.

phenotypes (negative for CD45RA, CD27, and CD62L), while Clusters 4 and 7 expressed higher levels of CD27 and low levels of CD45RA (Fig. 3b), suggesting these memory populations were less differentiated. Cells in Clusters 3 and 8 expressed CD127 and CD62L, suggesting a phenotype closer to that of $T_{CM}$ (Fig. 3b).

BCG revaccination induced significant increases in the proportion of CD4⁺CD69⁺CD154⁺ T cells in Clusters 5 and 9, whereas H4:IC31 vaccination increased the proportion of these cells in Clusters 4, 5, 6, and 9 (Fig. 5a). Neither vaccine changed the proportion of these cells in the $T_{CM}$ cluster (Cluster 3) nor in two of the less-differentiated $T_{EM}$ clusters (Clusters 7 and 8).

To gain insight into the clusters that increased post-vaccination, we compared the effector memory clusters to each other. Relative to the other effector memory clusters, cells in Cluster 4 expressed significantly higher levels of *IL2, TNF* and *IFNG* transcripts (Fig. 5c, d), genes encoding the activation markers *CD69, CD38, CD40L* and *CD137*, and the chemokine receptor *CCR4* (Supplementary Data 9), exhibiting a highly polyfunctional phenotype similar to that of cells in the high-abundance Cluster C identified via ICS.

Cluster 5, the largest of the $T_{EM}$ clusters, showed a strong Th1-type profile, with significantly higher expression of CD26 by ADT (Fig. 5b) and significantly higher levels of transcripts for *IFNG* as well as the cytotoxic molecules granzyme A (*GZMA*) and granulysin (*GNLY*) (Fig. 5c, d). Cluster 5 also highly expressed several homing-related transcripts relative to the other $T_{EM}$ populations: the chemokine receptor transcripts *CCR6, CXCR4, CCR2, CXCR6* and *CCR5* as

well as the cell trafficking molecule, *ITGB1*, suggesting these cells have increased migratory capabilities (Fig. 5c and Supplementary Data 9).

Cells in Cluster 6 expressed high levels of *GATA3*, the master regulator of Th2 T cell differentiation[42], as well as *IL13* (Fig. 5c, d), indicating that this cluster consisted of Th2-type CD4⁺ $T_{EM}$ cells.

Cells in Cluster 9 by ADT expressed high levels of CD103 (also known as integrin αE), a marker associated with tissue-resident T cells[15], and the chemokine receptor CXCR5 (CD185), known for its expression by T follicular helper cells and is upregulated upon activation (Supp Data 10)[43]. This cluster also expressed higher transcript levels of *CCR6* (CD196) as well as *CD226*, an adhesion molecule involved in cell migration, activation, proliferation and differentiation[44] (Supplementary Data 9). In addition, the genes *IL17F* and *IL22*, typically expressed by Th17 cells, were highly expressed in Cluster 9 relative to the other memory cell clusters (Fig. 5c, d). IL-17 and IL-22 are associated with maintenance of mucosal barrier homeostasis[45] and are found in the lungs of TB-infected individuals[15], suggesting that Cluster 9 may be circulating Th17 cells with the ability to home to tissues.

## No vaccine-induced effect on CD8⁺ T cell clusters

A single cell analysis of the 7952 CD8⁺ activated T cells sorted by expression of CD69 and CD137 identified 8 distinct clusters distinguished by ADT and gene expression (Supplementary Fig 8 and Supplementary Data 11, 12). No significant changes were observed post

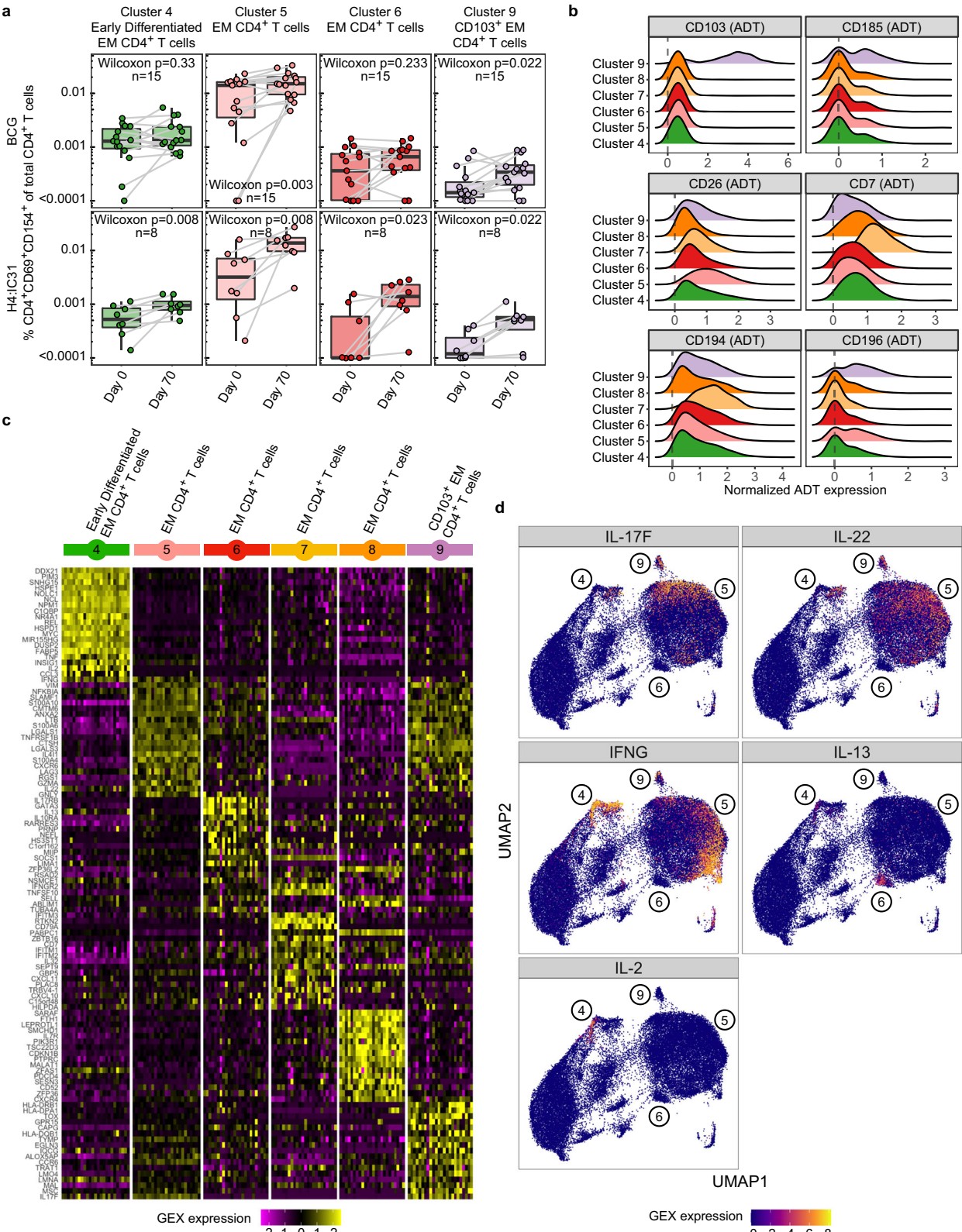

**Fig. 5 | TB vaccination boosts multiple T_EM clusters. a** Proportion of activated T cells of total CD4+ T cells, measured by CITE-Seq, present in each T_EM cluster that showed a significant change after vaccination in either the BCG ($n = 15$) or the H4:IC31 ($n = 8$) vaccine groups. *$p < 0.05$, **<0.01, two sided Wilcoxon Signed Rank test. Each point depicts the response in a single participant. Boxplots indicate the median response and the first and third quartiles; whiskers extend to no further than 1.5 times the inter-quartile range. **b** Ridge plot showing normalized expression values of antibody-derived tags (ADT) for selected surface markers for the T_EM CD4+ T cell clusters. **c** Heatmap showing average gene expression by participant (columns) of the top 20 genes (rows) that distinguish the 6 antigen-reactive T_EM CD4+ T cell clusters from each other. **d** Overlay showing cells expressing transcripts of selected cytokines on the UMAP projection of all activated CD4+ T cells. Cluster numbers are shown on the figure.

vaccination for any cluster and thus no further analyses were performed.

## M.tb-activated T cell receptor clonotypes

We next analyzed the T cell receptors (TCRs) sequenced in the CITE-Seq analysis to determine whether any clonotypes we identified had also previously been found in *M.tb*-reactive T cells from other published studies. We analyzed sequences from 37,405 T cells that expressed a paired-chain αβ TCR (see Methods for details). Only a handful of clonotypes were found in the repertoires of more than one individual (i.e., public TCRs), based on exact matching of the CDR3 amino acid sequence and VJ-gene usage. Therefore, we sought to identify public clusters of similar clonotypes by constructing a network of TCRs with similar CDR3 sequences, making it likely that they recognize the same antigen[46,47]. The data were then merged with two published datasets of TCRs sequenced from putatively *M.tb*-reactive T cells[48,49]. We found that 18.7% of the *M.tb*-reactive αβ TCRs in our study clustered with at least one other clonotype from another individual in one of the three datasets, providing further evidence that the activated sorted cells were *M.tb*-reactive (Supplementary Fig 9). The remaining TCRs did not cluster with TCRs from more than one individual, yet may still be *M.tb*-reactive; this low level of publicity is not unexpected given the enormous diversity of individual TCR repertoires, the diversity of participant HLA genotypes and the breadth of potential T cell epitopes in *M.tb*. To facilitate future identification of public responses, the TCR data is publicly available in the data repository associated with this manuscript (https://doi.org/10.6084/m9.figshare.24150492).

## Cytokine secretion increases after vaccination

To complement the analysis of antigen-induced proteins and genes in individual cells we examined the concentrations of 20 chemokines and cytokines in the PBMC supernatant following the 16-h antigen stimulation described above. Six analytes were significantly increased at day 70 among BCG recipients: IFN-γ, IL-17a, IL-17a/f, IL-2, IL-22, and lymphotoxin-α (LTα), (FDR $q < 0.2$; Supplementary Fig. 10, Supplementary Data 13). Notably, these increases were consistent with the broad transcriptional analysis of CD4$^+$ T cells which showed increases of transcripts for these analytes in the BCG group (e.g., *IFNG*, *IL17A*, and *IL22*), further emphasizing the induction of Th1 and Th17-type CD4$^+$ T cells.

Cells in the H4:IC31 group, like those from the BCG group, exhibited increased secretion of IFN-γ, IL-2, IL-17, IL-17f and LT-α at day 70 compared to baseline, consistent with the increases in the proportion of Th1 and Th17-type antigen-reactive cells observed by CITE-Seq. However, the H4:IC31 group differed in that there was not a significant change in secretion of IL-22, while there was a significant increase in secretion of IL-5 and CXCL10 (IP-10). This increased secretion of IL-5 further supports the induction of Th2 response by H4:IC31, consistent with the ICS assay data.

In summary, both BCG revaccination and H4:IC31 vaccination increased CD69$^+$154$^+$ Th1 and Th17 effector memory CD4$^+$ T cell populations. Consistent with previous reports on the IC31 adjuvant, the H4:IC31 vaccination also induced some Th2 cell responses.

## Discussion

The BCG revaccination regimen demonstrated potential to prevent sustained *M.tb* infection in the C-040-404 phase 2 prevention of infection trial[6,50]. Here, in adolescents who were previously vaccinated with BCG at birth and who live in a high TB burden region, we identified several previously undescribed subsets of CD4$^+$ T cells that were induced by BCG and H4:IC31 as boosters given 12–17 years after the prime with BCG and may contribute to the protection they conferred.

The use of both 26-color multiparameter flow cytometry and CITE-Seq provided further, complementary information about T cell

responses to revaccination with BCG or vaccination with H4:IC31. The ICS assay quantitatively evaluated antigen-reactive T cell subsets using defined cytokines expressed intracellularly and provided additional memory and functional marker cell surface expression following a relatively short stimulation (6 h) using a limited set of fluorescently-tagged antibodies. By contrast, CITE-Seq analyzed activated single cells after a 16 h antigen stimulation and permitted deep phenotypic interrogation of transcriptional signatures and surface markers using 108 oligo-tagged antibodies. Because the kinetics and regulation of RNA and protein expression differ by marker, some markers may be more easily detected in one assay versus the other. In addition, ex-vivo stimulation of the PBMC for CITE-Seq may have altered the transcriptomic profile of the cells, making the definition of cell populations based on gene expression alone less accurate. Nonetheless, when viewed together, the analyses presented provide a comprehensive immune assessment of the vaccine effects without long stimulation times and cell division, and they can be applied to high quality, previously cryopreserved PBMC for longitudinal studies.

Our immunogenicity studies coincided with a worldwide shortage of the Danish BCG vaccine (manufactured by the Statens Serum Institut, SSI) so we were unable to use the homologous BCG for stimulations in our assays; we instead used BCG Pasteur provided by the Aeras Foundation that sponsored the C-040-404 trial. Different strains of BCG have been shown to elicit distinct cellular response profiles[51,52], so we may have underestimated vaccine-induced responses here. In future studies, stimulation with the homologous BCG will hopefully be possible.

CITE-Seq enabled us to identify two naïve-like CD4$^+$ T cell clusters, one of which decreased after both BCG revaccination and H4:IC31 vaccination (Cluster 1). These findings suggest that vaccination may either drive CD4$^+$ T cells to differentiate into other sub-populations of antigen-reactive cells or initiate migration to peripheral tissues. We classified these clusters as naïve-like since they expressed canonical naïve cell surface memory proteins, were activated after a 16 h stimulation (enriched from cell sorting using our AIM assay markers, CD69 and CD154) and expressed distinct cytokine gene transcripts (*CXCL10* and *IL16*).

The heterogeneity of naïve-like CD4$^+$ T cell populations is an area of ongoing investigation. Such cells have been categorized as naïve receptor CD4$^+$ T cells (T$_{NR}$), stem cell memory T cells (T$_{SCM}$), and cytokine-producing naïve CD4$^+$ T cells (T$_{CNP}$)[35,53]. Directly fitting the subsets and clusters of naïve-like cells described in our study with the previously reported categories proved challenging, likely due to differences in the cohorts studied (e.g., infection vs. vaccination, participant age, previous mycobacterial exposure and initial BCG vaccination status) as well as to assay differences (e.g., stimulation versus identification of antigen-specific cells by tetramers and timing of analysis). For example, we looked for induction of T$_{SCM}$, which are characterized by the expression of CD95, CD58, CD11a, CXCR3 and IL-2RB[53], but did not find differential expression of these genes or proteins in either naïve-like cluster compared to other CD4$^+$ T cells in our study.

CITE-Seq enabled characterization of peripheral blood T$_{EM}$ subsets induced by TB vaccines with an unprecedented level of granularity. We identified four T$_{EM}$ clusters that increased after H4:IC31-vaccination (Clusters 4, 5, 6 and 9), two of which also increased after BCG revaccination (Clusters 5 and 9). Previous studies of BCG and H4:IC31 vaccination have described these antigen-reactive cells as one polyfunctional Th1 memory CD4$^+$ T cell subset and have mostly restricted their characterization to the analysis of IFN-γ, IL-2, TNF and IL-17[6–8,10,17,18]. Cluster 5, the cluster with the largest proportion of antigen-reactive CD4$^+$ T cells, expressed Th1 cytokines, cytotoxic molecules and chemokine receptors; the majority of the cells described in previous BCG and H4:IC31 vaccination studies likely belong to this cluster. BCG revaccination in the C-040-404 trial boosted antigen-

reactive CD4[+] T cells expressing IFN-γ and/or IL-2[6] as well as polyfunctional Th1 cells and antigen-reactive Th22 cells[9]. In the present study, we observed a BCG revaccination-mediated increase in Th1 cells via both ICS and CITE-Seq. The Th1* T cell population (CXCR3[+]CCR6[+]) has garnered attention in the field and while the exact role of this population is still not clear, these cells have been implicated in the control of *M.tb* infection[12,54], making their induction by vaccination an attractive strategy. Studies have shown the Th1* population makes up the majority of *M.tb*-reactive T cells after infection[12,54] and vaccination[13]. In this study we found that 26–41% of the antigen-reactive cells identified by ICS had a Th1* phenotype; the clusters containing these cells also increased after vaccination (Clusters B, C and I). We were unable to detect the Th1* subset of cells in our CITE-Seq analysis, likely due to the kinetics of expression of these proteins. Further studies are required to understand the role of these populations in providing protection against TB in humans.

By CITE-Seq we were also able to identify CD4[+] T cells expressing *IL22*, previously shown to be involved in anti-mycobacterial responses[9]. Expression of *IL22* was detected in effector memory Cluster 5 and *IL17F* and *IL22* in Cluster 9, which increased post-vaccination. The expression of cell migration markers and Th17-associated cytokines by the small Cluster 9 suggests that these cells are poised to respond to *M.tb* and may circulate into tissues such as the lung, where Th17 cells have been found in TB-infected individuals[15]. While we observed an increase in IL-17a, IL-17a/f and IL-22 secretion from PBMC, suggesting induction of a Th17 response, we did not identify a population of Th17 cells by ICS. We believe this may be due to the relatively short stimulation time (6 h) used in our ICS assay.

Future studies could also leverage the T cell receptor clonotypes we identified in this study that clustered with similar clonotypes derived from other repositories of *M.tb*-reactive T cells; these public TCRs may represent dominant *M.tb* or BCG T cell epitopes that could be useful vaccine targets.

A limitation of this study is its dependence on the observation of changes in the peripheral blood. Recent non-human primate (NHP) studies of BCG vaccination are providing insights into T cell populations in the lung that may be important immune correlates of protection. Intravenous (IV) BCG vaccination of NHP was highly protective against *M.tb* infection and the largest proportion of antigen-reactive cells post BCG vaccination were Th1 cells[6,9], with a smaller proportion (<10%) co-expressing Th1 cytokines and IL-17 (named Th1/17 cells)[8]. Subsequent work has demonstrated that these CD4[+] T cell populations in the lungs correlate more strongly with protection than those identified in the peripheral blood[55] and that transcriptional signatures of lung CD4[+] T cells may be important for bacterial clearance. In our study, Cluster 4 shared many of the features of this Th1/17 cell population, with only approximately 3% of the cells co-expressing *IFNG* and *IL17F* transcripts, despite the difference in the route of BCG administration in our human study. Darrah et al.[8] also identified cells expressing CD154 but not Th1 or Th17 cytokines, consistent with Cluster D in our ICS data, which was significantly boosted after BCG revaccination. Cluster D was present at relatively high frequencies, but did not express any cytokines that we measured, suggesting a need to further identify the functional features of those cells and how they contribute to the vaccine response. In addition, T cells co-expressing CD69 with CD103, a marker expressed by some tissue resident memory (T_RM) T cells, were present in the NHP lung parenchyma and likely represent tissue resident T cells. Cells in Cluster 9 of our CITE-Seq data shared features with this population of cells, expressing high levels of CD103 protein and overexpressing *CCR6* relative to the other T_EM populations, which may indicate that intradermal BCG vaccination may prime tissue-homing antigen-specific T cells.

The present study focused on conventional CD4[+] T cell responses to TB vaccination, as CD8[+] T cell responses in our assays were infrequent and low in magnitude, consistent with other BCG revaccination

and H4:IC31 vaccination studies[9]. The lack of antigen-reactive CD8[+] T cells detected in our study compared with studies done in other cohorts[56–58] is likely due to differences in assay set-up: we used shorter stimulation times, lower peptide concentrations and did not make use of antigen-pulsed feeder cells. In addition, *M.tb* specific CD8[+] T cell response frequencies have been shown to be higher in sensitized individuals and in individuals with active TB disease as compared to the QFT negative adolescents in our vaccine trial[58]. The roles of donor-unrestricted T cell (DURT) populations, such as NKT cells, mucosal-associated invariant T cells (MAITs) and γδ T cells, are also being investigated in response to BCG vaccination. While our results focus on the CD4[+] T cells, we have provided open access to single-cell resolution data of many other putatively antigen-reactive T cells from the CITE-Seq analyses that deserve further evaluation in future studies.

Together, our data show that both BCG and H4:IC31 vaccination increased a large TB-reactive Th1 effector memory cell subset and a smaller Th17 cluster of cells that, based on their expression of CD103, may represent recirculating T_RM cells migrating to and/or exiting from lung. The data also highlight heterogeneity in effector memory subsets and the importance of integrated analysis of gene and protein expression to understand T cell biology and the cell subsets that contribute to vaccine response. Although the present study of T cells is from the phase 1b HVTN 602 trial and is limited to an immunogenicity assessment, our findings revealed unique clusters of both naïve-like and memory CD4[+] T cells that will be interrogated in the planned studies to identify the correlates of protection in the C-040-404 prevention of infection trial, as well as in other studies of BCG and recombinant mycobacterial vaccines.

## Methods
### HVTN 602 trial
The HVTN 602/Aeras A-042 phase 1b study was conducted at the Emavundleni Clinical Research Site (CRS) in Crossroads, Cape Town, South Africa[10]. The study was approved by the University of Cape Town Ethics Committee. The HVTN 602 trial is registered with the U.S. National Institutes of Health Clinical Trials Registry (ClinicalTrials.gov identifier NCT02378207). Participants provided written informed assent prior to inclusion in the study and legal guardians/parents provided written informed consent[10].

Healthy, HIV and QuantiFERON-negative adolescents aged 12–17 years who had received BCG vaccination (SSI, $2–8 × 10^5$ cfu) at birth were enrolled in the trial and were randomized to receive placebo, 1 dose of BCG (SSI, $2–8 × 10^5$ cfu in 0.1 ml administered intra-dermally) or 2 doses of the H56:IC31 or H4:IC31 vaccines administered intra-muscularly (days 0 and 56; participants in the H56:IC31 arm were not included in the present study). The demographics of participants from whom samples were assayed in this study are included as Supplementary Data 14.

### Intracellular cytokine staining (ICS) assay
T cell responses to mycobacterial antigens were measured by intracellular cytokine staining using multiparameter flow cytometry as previously described[59,60] in cryopreserved PBMC from blood collected at study days 0, 70 and 168, as well as day 14 (H4:IC31 group) and day 28 (BCG group). Cryopreserved PBMC were not available for all participants at all timepoints due to some missed clinic visits or low-volume blood draws. Briefly, cryopreserved PBMC were thawed, incubated overnight and stimulated on day 2 for 6 h at 37 °C with either peptide pools (peptides of 15 amino acids overlapping in sequence by 11 amino acids) for the vaccine-matched proteins (Ag85B and TB10.4, final concentration of 1 µg/ml in DMSO, Bio-Synthesis Inc.), BCG (Pasteur strain from glycerol stocks at a final MOI of 9 [provided by Aeras]), gamma-irradiated *M.tb* H37Rv whole cell lysate (BEI resources, cat NR-14822, final concentration of 30 µg/ml), dimethyl sulfoxide (DMSO, 0.5%, Sigma Aldrich; negative control) or staphylococcal

enterotoxin B (SEB, 0.25 μg/ml; Sigma Aldrich; positive control) in the presence of costimulatory antibodies CD28 and CD49d (1 μg/ml, BD Biosciences) and brefeldin A (BFA, 10 μg/ml, Sigma Aldrich). Cells were incubated with EDTA (2 mM, Life Technologies) overnight at 4 °C, then stained with a 26-color antibody staining panel (modified version of[51], CD26 fluorochrome changed to BUV396, CD56 BB780 added, CD4 fluorochrome changed to BUV496, CD45RA changed to BV711, CXCR3 fluorochrome changed to PE-Cy7 and CD154 fluorochrome changed to PE-Cy5; Supplementary Data 15). The stained cells were then acquired on a BD FACSymphony A5 flow cytometer (BD Biosciences) using BD FACS Diva version (v) 8.0.1 and analyzed using FlowJo v9.9.6 (FlowJo LLC).

### Computational analyses of ICS data

To assess the response of CD4+ and CD8+ T cells to each antigen stimulation ex vivo, antigen-reactive cells were defined as cells expressing at least 1 of the following markers: IL-2, IFN-γ, TNF, IL-17a, IL-4/IL-13, IL-22 or CD154. The magnitude of the response to an antigen was computed as the percentage of total CD4+ or CD8+ T cells expressing at least one of the cytokines minus the same percentage measured in the negative control condition.

The fluorescence intensity of all the markers included in the ICS panel (except CD3, CD4, CD14, CD16 and CD56) on CD4+ T cells expressing at least 1 of 7 markers (IL-2, IFN-γ, TNF, CD154, IL-17a, IL-4/13 or IL-22) was extracted and used as input for UMAP visualization as well as for clustering algorithms. Visualization in high-dimensional space was performed with UMAP using the R package uwot (v0.1.16) with the following parameters: *n_neighbors = 10* and *min_dist = .1*. The PhenoGraph algorithm[29] was used for clustering. The Rphenograph R function from the Rphenograph R package (v 0.99.1) was computed with the default parameters. Clusters were annotated according to their marker expression profiles. Marker expression across different clusters was assessed using heatmaps and gates defined by manual gating in FlowJo. Hierarchical clustering was used to manually merge correlated clusters; Pearson's correlation was computed among cluster expression profiles and the clustering dendrogram was cut to yield 20 individual clusters.

Quantification and background subtraction of reactive cell frequencies was calculated for each cluster[62]. Wilcoxon signed rank tests between baseline (day 0) and post-vaccination days (day 14/28, 70, 168) were used to identify clusters with statistically significant differences induced by vaccination. Multiplicity adjustment was made over the number of clusters within each arm, timepoint and stimulation, FDR $q < 0.05$ was considered significant.

### Enrichment of T cells via the AIM assay for CITE-Seq

Cryopreserved PBMC were thawed and resuspended in 5 ml RPMI-HEPES (R10 HS, Gibco/ThermoFisher) supplemented with L-glutamine and penicillin-streptomycin (Gibco/ThermoFisher) containing 10% human AB serum (Gemini Bio) and incubated at 37 °C/5% CO$_2$ for 3 h. Cells were counted and resuspended in R10 HS containing 1 μg/ml anti-CD40 antibody (Miltenyi) and anti-human CD28/49d (BD Biosciences) at a concentration of $5 \times 10^6$ cells/ml. Specimens from all visits for each participant were run on the same day to minimize batch effects. A minimum of one million cells were plated for each stimulation condition. Those from the BCG group were stimulated with gamma-irradiated *M.tb* H37Rv whole cell lysate (BEI resources, cat NR-14822, final concentration of 30 μg/ml), or PBS (Gibco/ThermoFisher; negative control), whereas those from the H4:IC31 group were stimulated with the Ag85B/TB10.4 combined peptide pool (concentration 1 μg/ml of each peptide, Biosynthesis Inc.) or DMSO (0.5%, Sigma Aldrich; negative control), and incubated at 37 °C/5% CO$_2$ for 16 h. At the end of the stimulation period, cells were pelleted and 150 μl of supernatant was removed and stored at −80 °C for measurement of soluble proteins.

A master pool containing 108 oligo-labeled antibodies – Ab-derived tags, or ADTs, (Supplementary Data 16) was made one day prior to starting the sorts to minimize batch effects. On the day of each sort, aliquots (sample pools) of the master pool were then spiked with a prespecified Hashtag oligo antibody pool (HTO, Biolegend), an in-house oligo labeled CD153, which was made per the manufacturer's recommendations using Thunderlink PLUS oligo Antibody conjugation kit (NovusBio), and the fluorescently labeled antibodies (Supplementary Data 17). A custom batch of the oligo-labeled tetramer MR1-5-OP-RU was made using biotinylated MR1-5-OP-RU monomers (NIH Tetramer Core Facility) and a streptavidin-oligo labeled reagent (SRA-O, Biolegend) per the manufacturer's recommendations and was used to stain the cells sequentially prior to adding the sample antibody pool.

The cells were washed and stained with Live/Dead Aqua viability dye for 30 min, washed with Cell Staining Buffer (CSB, Biolegend) supplemented with Human TruStain FcX (BioLegend) (FcX CSB) and then stained with an aliquot of the oligo labeled MR1-5-OP-RU for 15 min, cells were washed with CSB, and then stained with the aliquot of the sample pool antibody cocktail containing fluorochrome labeled (Supplementary Data 17) and oligo labeled antibodies (Supplementary Data 16), and prespecified HTO, prepared in FcX CSB for 30 min. Cells were washed and resuspended to $5 \times 10^6$ cells/ml in FcX CSB containing 200 U/ml RNasin (ThermoFisher) and filtered using a 40-micron filter-topped FACS tubes (Corning) prior to sorting. Cells were sorted on the FAC-SAria II SORP (BD Biosciences) equipped with DIVA 8.0.1 software. The following populations were sorted into 2 ml LoBind tubes (Eppendorf) containing 20 μl FcX CSB 200 μ/ml RNasin: CD3+CD4+CD154+CD69+ and CD3+CD8+CD69+CD137+ T cells (antigen-reactive CD4+ and CD8+ T cells), non-antigen-reactive T cells (CD3+CD4+ but not CD154+, or CD3+CD8+ but not CD137+), and CD3− cells from each stimulation condition (Supplementary Fig. 2b, c).

### 10X Genomics immune profiling with VDJ and Feature barcoding

Captured cells were pooled according to a predetermined pooling and hashtag schema (Supplementary Data 18) and then processed using the 10X Genomics 5′ immune profiling with VDJ using the kits listed in Supplementary Data 19 and feature barcoding Chromium version 1.1 workflow generating cDNA, Feature barcode (including hashtag), VDJ (TCR α/β) libraries that were indexed and sequenced using the Illumina Novaseq (Illumina) sequencing platform using S4 200 flow cells per the manufacturer's instructions. Gene expression (GEX) libraries were sequenced at a minimum of 20,000 reads per cell, feature barcode libraries and the TCR α/β libraries at a minimum of 5000 reads per cell. All sequencing libraries were QC'd using the Agilent 4150 TapeStation system (Agilent) and for the GEX and TCR libraries using the high sensitivity D5000 ScreenTape and reagents, for the FBC libraries the D1000 ScreenTape and reagents were used per the manufacturer's recommendations.

### Pre-processing of sequencing data

The steps for data processing, quality control and analysis are outlined in Supplementary Fig. 3. Raw base call (.BCL) files were used as input to the Cell Ranger Single-Cell Software (version 3.1.0) using the *count* and *vdj* options to perform sample demultiplexing, barcode processing and single-cell 5′ feature counting. Reads, containing cDNA inserts, were aligned to the hg38 human reference genome using STAR. Aligned reads were then filtered for valid cell barcodes and unique molecular identifiers (UMIs). Cell barcodes with 1-Hamming-distance from a list of known barcodes were examined. UMIs with sequencing quality score >10% and not homopolymers were retained as valid UMIs. A UMI with 1-Hamming-distance from another UMI with more reads, for a same cell and a same gene, was corrected to this UMI with more reads. Read metrics were analyzed from the generated metric summary file to ensure all libraries had sufficient coverage.

## Demultiplexing with hashtag oligos

The Seurat package (version 4.0.1), HTODEMUX was used to demultiplex these sequences using the unique oligo barcodes to demultiplex cells to their original sample of origin and to filter any cells that may contain cross sample "doublets"[63,64].

## Data analysis for CITE-seq

Quality control for the data was performed as follows: following sequence alignment and demultiplexing, cells with high (>5000) feature counts (as proxy for exclusion of droplets containing more than one cell) or very low (less than 200) (as proxy for exclusion of empty droplets) were excluded[65]. Cells with high mitochondrial proportions (as proxy for cell damage) were also filtered out. For each combination of batch and pool, a threshold of 5 Median Absolute Deviation (MAD) for percent of mitochondrial genes was used. Cells with ADT counts were retained; whereas cells with unusual low numbers of detected ADTs, defined here as half of the median across all cells for each combination of batch and pool, were removed (57 cells were filtered out at this step). As above, we also used a generous cutoff (5 MADs) to exclude cells with large ADT counts (as proxy for "sticky" cells). After filtering, all samples were merged, resulting in a dataset of 241,428 cells for downstream analyses.

To analyze the data for the antigen-reactive & non antigen reactive cells, standard analyses were performed using the Seurat R package (v4.0.1). We first integrated the dataset using sample (combination of each batch, PTID, arm, stimulation and visit) as factor with the anchor-based workflow implemented within Seurat[64].

To integrate the gene expression values, we separately normalized each of our 64 partitions using the SCTransform R function[66]. Data integration was performed using the FindIntegrationAnchors and IntegrateData functions (with *dims = 1:50*, *reduction = rpca* and *reference = 1:2* corresponding to the first participant ID & batch)[64]. Principal component analysis (PCA) on the integrated dataset was then performed using the RunPCA R function. The top 50 principal components (PCs) were used for UMAP visualization and the weighted nearest neighbor integration. We also integrated the ADTs across samples. For each of our partitions, a centered log ratio (CLR) transformation was performed (the NormalizeData function with *normalization.method = "CLR"*). Data integration and visualization were then performed using the same workflow as described above.

The weighted nearest neighbor (WNN) workflow was investigated. As described in[67], this workflow consists of three steps: (1) independent preprocessing and dimensional reduction of each modality individually; (2) learning cell-specific modality *weights* and constructing a WNN graph that integrates the modalities; and (3) downstream analysis of the WNN graph. The FindMultiModalNeighbors R function was computed with *dims.list = list(1:50, 1:50)*, *prune.SNN = 1/20* and *modality.weight.name = c("RNA.weight", "ADT.weight")*. UMAP with these parameters (*min.dist = .01* and *nn.name = "weighted.nn"*) was performed using the RunUMAP function. Cell clustering was then performed using the FindClusters function implemented in Seurat (*graph.name = "wsnn"*, *algorithm = 3* and *n.start = 10*) with different resolutions.

To analyze the antigen-reactive CD4⁺ T cells, two ADTs (CD4 and CD8) were used to determine CD4+ and CD8 + T cells using the tsGates function of the SCAMP R package[68]. Antigen-reactive CD4 + T cells were selected and the WNN workflow was re-run as above.

First, PCAs for both gene expression and ADT expressions were re-computed. Then, the FindMultiModalNeighbors R function was run with *dims.list = list(1:50, 1:25)*, *prune.SNN = 1/20* and *modality.weight.name = c("RNA.weight", "ADT.weight")*. UMAP with these parameters (*min.dist = .1*, *n.neighbors = 15* and *nn.name = "weighted.nn"*) was performed using the RunUMAP function. Cell clustering was performed as above using the FindClusters function implemented

in Seurat (*graph.name = "wsnn"*, *algorithm = 3* and *n.start = 10*) with a resolution of 1.

To identify gene cluster markers, the FindAllMarkers R function was used with *test.use = MAST*, *latent.vars = c("nCount_RNA", "batch", "ptid", "arm", "stimulation", "visit")*, and *max.cells.per.ident = 5000*. To identify ADT cluster markers, the same *FindAllMarkers* R function was used with *test.use = LR*, *latent.vars = c("batch", "ptid", "arm", "stimulation", "visit")*, and *max.cells.per.ident = 5000*. We merged clusters that did not exhibit clear evidence of separation based on gene and ADT markers, as well as UMAP embeddings. At the end, twelve clusters were identified (1: Naive-like CD4⁺ T, 2: Naive-like CD4⁺ T, 3: CM CD4⁺ T, 4: Early Differentiated EM CD4⁺ T, 5: EM CD4⁺ T, 6: EM CD4⁺ T, 7: EM CD4⁺ T, 8: EM CD4⁺ T, 9: CD103⁺ EM CD4⁺ T, 10: Regulatory T, 11: NKT & γδT and 12: ADT-high). Cluster 12, in which we detected cells with higher number of ADTs than in the other clusters, was removed for Figures and downstream analyses as it might be due to unspecific binding.

Differential expression analysis was performed using the R package MAST (v0.1.1.16)[69]. Differential expression was performed between post-vaccination days. The normalized gene-cell barcode matrix was used as input. The model included the batch and participant ID as covariates, as well as the cellular detection rate (CDR), defined as the total of UMIs in a given cell. Genes were declared significantly differentially expressed at a false discovery rate (FDR) of 5% and a fold-change >1.5.

## T cell receptor sequence analysis from BCG vaccinees

TCR data for single cells was generated from the scPGP 10x workflow using the V(D)J 5′ reagent kit (Supplementary Data 19). Data was processed using CellRanger (v3.1.0), with additional processing to remove highly expressed chains representing contamination. For the clustering analysis, we used gene expression clusters to identify T cells in the antigen-stimulated condition and analyzed cells with a complete αβ paired TCR (*n* = 37,405 clonotypes from 16 BCG recipients). Data from this study was then combined with data from two studies that also sorted M.tb-reactive T cells: Huang et al. (n = 2608) clonotypes from 24 individuals[48] and Musvosvi et al. (*n* = 16,253 clonotypes from 70 individuals)[49]. Clonotypes from all three datasets were clustered by first computing all pairwise distances using tcrdist3[47] and generating a network with edges connecting all clonotypes with a TCRdist <100 tdus (approximately 2–3 amino acid substitutions in the CDR3 loops in each chain); for paired-chain analysis this threshold was previously found to be highly specific for grouping TCRs that recognize the same antigen. Public clusters were defined as those containing clonotypes from more than one individual. Network plots were created for public clusters with at least five members.

## Multiplex cytokine analysis

Frozen aliquots of supernatants from PBMC stimulated for 16 h with gamma-irradiated *M.tb* H37Rv whole cell lysate for the BCG group (BEI resources, cat NR-14822, final concentration of 30 μg/ml), whereas those from the H4:IC31 group were stimulated with the Ag85B/TB10.4 combined peptide pool (concentration 1 μg/ml of each peptide, Biosynthesis Inc.), both groups with lipopolysaccharide-LPS (Invivogen; concentration of 10 ng/ml; positive control), or PBS (Gibco/Thermo-Fisher; negative control) were thawed at room temperature and 20 secreted cytokines and other factors present in PBMC were analyzed using the following custom Human Biomarker kit (catalog no. K151A9H-2) from Meso Scale Discovery: Proinflammatory Panel 1 (IFN-γ, IL-1β, IL-2, IL-6, IL-8, IL-10, IL-13, TNF), Cytokine Panel 1 (GM-CSF, IL-5, IL-12/23 p40, IL-17A, LTα), Chemokine Panel 1 (MIP-1β, IP-10, MCP-1, MDC), Th17 Panel 1 (IL-21, IL-22), and Cytokine Panel 2 (IL-17A/F), run according to the manufacturer's instructions on the MESO QuickPlex SQ 120 platform (Meso Scale Discovery). Data were analyzed using the nCal R package[70]. Changes in cytokines were defined as the log fold-change over pre-vaccine levels post vaccination. Wilcoxon signed rank

tests were used to determine significant changes (two-sided). For each arm and stimulus, multiple comparison adjustment was performed across analytes using the Benjamini & Hochberg method. For this assay we selected an FDR $q < 0.20$ for reporting significance, as the primary purpose of the assay was to generate hypotheses for future follow-up testing in a larger study.

## Reporting summary
Further information on research design is available in the Nature Portfolio Reporting Summary linked to this article.

## Data availability
All data supporting this study have been deposited in an archive on Figshare and are publicly available at this hyperlink: https://figshare.com/articles/dataset/HVTN_602_data_repo_Adolescent_BCG_revaccination_induces_a_phenotypic_shift_in_CD4_T_cell_responses_to_Mycobacterium_tuberculosis/24150492. The raw transcriptomic data are submitted to the BioProject website with project number PRJNA1095450 https://www.ncbi.nlm.nih.gov/bioproject/1095450 and available from the corresponding authors upon request. The processed transcriptomics data are provided as aggregated gene counts per cell in the archive above. The raw FCS files for the intracellular cytokine staining assay are available at FlowRepository at the ID: FR-FCM-Z752. All raw data mentioned above, excluding raw data restricted by privacy laws, as well as other data are available in the article and its Supplementary files or from the corresponding author upon request. Source data are available with this paper. Source data are provided with this paper.

## Code availability
Newly-developed code used to analyze and visualize primary data are available at https://github.com/ValentinVoillet/BCG_revax (https://doi.org/10.5281/zenodo.10362451).[71]

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

## Acknowledgements

We thank all of the volunteers who participated in the HVTN 602/Aeras A-042 study, as well as the protocol team and clinical staff at the study sites. We acknowledge Dalene de Swardt, Shamiska Rohith, Stanley Loots, Ellen Shrontz, Stephany Wilcox and Margaret Mazyambe at CHIL for conduct of the ICS assays; Austin Varni, Sarah Byron, Gabriel Sanchez, Elisa Cano and Artem Yaschenko for assistance in performing the scPGP assays at FHCC; and Zinhle Mgaga at CHIL for performing the multiplex protein detection assay. We also acknowledge Gabe Murphy and Stephen Voght at FHCC for critical reading and feedback on the manuscript. We thank Thomas Scriba and Elisa Nemes for advice on markers to include in the panel. Research reported in this publication was supported by the National Institute of Allergy and Infectious Diseases of the National Institutes of Health under award numbers UM1 AI068618 (M.J.M.) and UM1 AI068635 (A.F.G.). The content is solely the responsibility of the authors and does not necessarily represent the official views of the National Institutes of Health.

## Author contributions

Conceptualization: O.B.D., E.A.N., M.J.M., S.C.D., J.G.K., L.G.B. Methodology: A.S., M.J.M., A.N., S.O., O.B.D., L.B.F., J.M. Visualization: O.B.D., V.V., E.A.N., A.F.G., E.W.N., L.B.F., S.C.D. Funding acquisition: M.J.M., E.A.N. Writing – original draft: O.B.D., E.A.N., V.V., S.C.D. Writing – review & editing: O.B.D., E.A.N., M.J.M., A.F.G., E.W.N.

## Competing interests

The authors declare no competing interests.
