## [Peer Review File · Nature Communications]

Adolescent BCG revaccination induces a phenotypic shift in CD4+ T cell responses to Mycobacterium tuberculosisREVIEWER COMMENTS

Reviewer #1 (Remarks to the Author):

This manuscript looks at the proteomic and transcriptomic profile of cells taken a separate immunogenicity study which was conducted in parallel with the POI study reported by Nemes et al NEJM 2018. A signal of protection against sustained QFT conversation was observed in the POI study for the BCG revaccination arm but not the H4/IC31 arm. The authors identify five effector memory CD4+ T-cell subpopulations associated with protection, together with a population of CD103 expressing Th17 cells.

The key limitation of this work is that the samples were taken from an immunogenicity study and there is no link or association with efficacy. The protection seen with BCG revaccination was only 45%, meaning there were also cases of sustained infection in the BCG revaccination arm. Given the heterogeneity in response seen here, it is difficult to associate any immunological signal with the protection conferred in the separate study. A second limitation of this work is that the H4 / IC31 boost vaccination regimen did not confer significant protection against sustained infection in this study. It is not possible to distinguish between whether that was because of statistical power (i.e. the trend would have been significant with larger numbers) or whether there is no effect of this vaccine. This confounds the analysis of the H4 samples.

Neither of these two significant limitations are discussed in this manuscript.

Overall this is an interesting descriptive study presenting a detailed immunological analysis of samples from this separate clinical trial. Without an individual association with protection, the conclusions that the markers identified are correlates of protection is a bit premature.

Specific comments:

One sentence summary: should be vaccination regimen not vaccine as is BCG revaccination.

Line 52 – morbidity due to TB disease – not infection

Line 59 – do the authors mean infection here? Again I suspect disease is what is meant.

Line 110-138 – the text describing Figure 1 does not make clear which results are statistically significant and which are not. Levels of statistical significance should be indicated in the text – for peak and also duration of response. Also, it is important to know if there are any differences between the 3 groups in Figure 1 – at the moment only within

group analysis is presented. Again statistical significance of comparison between groups is needed.

Line 150-164 – it would have been interesting to see the effector memory populations stimulated with BCG /Mtb lysate after H4 immunisation to allow direct comparison between groups.

Line 350 – it is not accurate to say the H4 / IC31 vaccination regimen showed potential to reduce sustained infection – this result was not statistically significant and the lack of efficacy may be real.

Reviewer #2 (Remarks to the Author):

- Knowledge of key determinates of immunological memory underlying vaccine effectiveness is of paramount importance for the fight against infectious diseases. This work characterizes PBMC samples collected from vaccine trial participants in order to discover biomarkers and immune cell characteristics that correlate with antigen recognition, particularly CD4+ T cells which have the capacity to effect immunological memory and associated disease resistance.
- To extensively characterize targeted subsets of PBMCs, the authors utilize single-cell analysis techniques: antibody-based multiplex protein profiling by flow cytometry; and additionally, CITE-seq which measures levels of a pre-determined panel of proteins as well as the transcriptome abundance profile of between 200 and 20,000 genes at the RNA level, for each individual cell. CITE-seq data, especially the RNA measurements are exceedingly complicated by the lack of uniform quality and completeness. A cogent transcriptomic profile, traditionally used to describe the phenotypic characteristics of a biological sample, is in single cell analysis dependent on the identification of groups of cells with relatively similar expression profiles. So while theoretically a complete protein expression profile and transcriptome are measured for each cell, in practice the data is significantly affected by random drop-outs and low dynamic range. Sophisticated statistical algorithms and best practices regarding data filtering and adjustments and calibrations, yet the application and tuning of these methods is unavoidably context-dependent and subjective. So it is important to make clear these decisions and the impact they have on the results and biological

interpretation of the results.

- The authors key claim is that the post-vaccination CD4+ T cell population is qualitatively different from the subject-matched pre-vaccination. This work builds on published work by transcriptional profiling of the CD4+ cells, specifically the subset capable of expressing activation markers in response to stimulation with vaccine antigens or a pathogen (Mtb) digest.

Major issues:

- Datasets are not accessible. Code/Scripts that could be inspected or used to re-run that analysis are not provided. Analysis workflow is complex enough to deserve a figure, pointing out subjective tuning steps (eg cluster consolidation decisions, relative RNA and ADT “weighting” applied in the weighted nearest neighbors, WNN, clustering).
- The transcriptome profile would be expected to respond more quickly to ex-vivo treatment conditions than would the protein expression profile. I don’t think it is made explicitly clear that the ex-vivo stimulation has potential complication of significantly changing the transcription profile of cells. Incubation with antigen proteins or Mtb lysate for 16 hr. Genes that have a functional role in T cell activation signaling, genes that are transcriptionally regulated in response to other cells being activated or responding to treatment components. Mtb lysate may have cytotoxic or other biological activity in addition to antigens protein, potentially affecting the transcription profile. All this makes the nomenclature “antigen specific” “antigen reactive” problematic in the context of transcriptome profiling (if not protein expression). An alternative might be the term “activated” cells? And where possible the functional definition of “CD4+CD154+ subset post-stimulation”?
- Line377: The kinetics and regulation differences impact not just assay sensitivity but the accuracy of gene function inferences.
- For a general audience at least, I think it would be helpful to be more systematic in the nomenclature referring to groups of cells. In vivo they might be “populations” by FACS sorting they might be (in vitro) “subsets” and in the context of proteogenomic profiling, an operational definition, (in silico) “clusters” could be used. If a novel cell type is “revealed” or “identified” by the expression profiles based clustering algorithm or heuristic, and not

supported by independent characterization, indeed it would be helpful to avoid confusion of the two. Characterization of clusters could be used to make inferences about subsets. Use subset characterizations to make inferences about in vivo populations of cells.

- The descriptions of the hypothetical subsets represented by Clusters is hard to follow because the evidence (key genes) they refer to is buried in the supporting tables. It would be helpful to have for each cluster an executive summary listing the few key genes described for biological interpretation and the cell type assigned. The key (for gene's RNA expression and/or protein primary data plotted across cohorts and time points. Expression levels and percentage cells positive can be indicated by y-scale, color, size of dot, etc. as needed. Short of biological validation, at least the verification using minimally processed data is needed in this case to depict the weight of empirical evidence for the hypothesized subset of T cells. This would lessen the burden on the technical performance of the CITE-seq data and fidelity of the data processing to support the inference from in silico cluster to potential FACS accessible subset and to pre-activation population of cells in study subjects.

- While it is reasonable to expect that the expression of proteins carefully selected to be informative with respect to known cell lineage and functions (i.e. marker proteins) would be consistent and complement the RNA expression profile, and effectively make the identity of individual cells more obvious, there is an informatic hazard that can occur when combining data from the two assays - RNA amplicons and protein ADT tags. In the organization and interpretation of clusters, the protein markers, because they are more reliably/consistently measured and easy to interpret relative to noisy RNA expression, could drive the clustering – making an “unsupervised clustering” operation insidiously supervised by the influence of domain information. I would like this to support my point that the information content of the protein and RNA data should be assessed separately, as is often the practice, carefully examine their concordance, and judiciously combine them in the analysis to understand the extent to which the clusters are driven by protein or RNA expression differences of key genes.

Minor:

- The manuscript could be improved by inclusion of more data quality diagnostics such as

protein/rna correlations of key genes, distribution of number of effector genes expressed (not just 1 to 7). Such exhibits would inform the reader the quality of the data and its characteristics to aid in re-use or comparison to other studies. Line 174 is very interesting but I could not find the promised details in the methods section.

- Terminology suggestion: Probably the “size of the cluster X” or the “proportion of cells matching the cluster X” would be more clear than the “abundance of cluster X”.
- Figures labels and data are not readable due in cases to poor contrast or impossibly small typeface. Proportions are sometimes reported as % and sometimes as fractions so where the y axis values labels or data reach only to 1,
- MAST differential expression outputs reported in supp tables do not always have p-value and adjusted p-value. They also propagate the error where the default outputs from the software label a fractional quantity as a percent ‘pct’. One can change the description in the table description to provide the correction.
- I believe the proper nomenclature for gene expression, as measured by RNA abundance, is the uppercase non-italicized. Referring to the GENE and not a transcript. The manuscript has lower-case italics.
- T cell receptor sequencing v(d)j analysis was mentioned but not analyzed or data provided.
- CD8+ subset CITE-seq analysis is not presented or data provided.
- L627 how many cells were dropped due to low ADT counts? What was the range exactly?
- L634 Please include analysis code or post on github.
- L656 Please provide more details about criteria for deciding the number of clusters were optimal for the dataset
- L687 fold change, log fold change, base 10 or base2, pFDR, FDR p-value, FDRq value – these terms are inconsistently used in the manuscript – It is hard to determine if these are typos. 5% or 0.05. Also the use of FDR at different significance level cutoffs, for 20 tests. Theoretically FDR for the top most significant result (of all tests) is just the p-value?
- L1142 Color scale legends for heatmaps are sometimes missing numeric values.
- Supp Tables have “Excel mutated” gene names appearing – apparently upstream corruption because gene descriptions are missing for those non-gene gene names.
- Supp tables are sometimes missing raw p-value or adjusted p-value
- L1064 What was the rate of detection of each of the effector genes?

- Supp figures L42 Typo, should be CD8+CD69+CD137+ ?

Reviewer #3 (Remarks to the Author):

In this submission by Dintwe, et al., the authors extend their initial report of the HVTN 602/Aeras A-042 trial (EClinical Medicine, 2020) to better characterize the CD4+ T cell response in South African adolescents to BCG revaccination or boosting with H4:IC31. The experimental approach is sound, using both multiparameter flow cytometry and CITE-Seq, as well as appropriate data analytics. Unsupervised cluster analysis of flow cytometric data revealed several CD4+ TEM populations that emerged especially after BCG revaccinations, two of which were polyfunctional, including one with Th1* features. Both vaccine regimens increased ag-specific TEM cells and CITE-Seq revealed several distinct TEM populations, including some with migratory potential and a small cluster of Th17+ cells that expressed high levels of CD103, suggesting these cells may represent TRM cells, shown in NHP models to correlate with protection in the lung. The manuscript has several strengths:

1. It carefully and comprehensively reveals the heterogenous CD4+ T cell response to these vaccines. While it is impossible to correlate specific populations with protection in this study, this work does provide valuable data that can be integrated with data from other TB vaccine trials.
2. Kudos to the authors for not just focusing on the cells themselves, but also measuring cyto/chemokines in supes, which provided confirmation of gene expression analyses (e.g. IFN-g).
3. Care must be taken when interpreting UMAP plots and WNN analysis, and the authors take care not over-interpret clusters consisting of few cells.

There are a few issues that limit the significance of the current manuscript.

1. As with most vaccine trials, the samples consisted of only cryopreserved PBMC; no airway cells (i.e. BAL) were collected. The authors do clearly acknowledge this limitation in the discussion.
2. There was no responses in CD8 T cells detected so these were excluded from further analysis. It is likely that the vaccine regimens did indeed induce a CD8 response as shown by

other studies of TB vaccines in humans. As acknowledged in the discussion, the failure to detect such a response here was likely due to assays conditions that favor CD4+, rather than CD8+, responses, but still represents a missed opportunity.

3. Subjects in the revaccination group received BCG SSI. But BCG Pasteur was used for in vitro restimulation here (as well as in the Eclin Med paper). Why was BCG Danish not used for restimulation? Is it reasonable to expect ex vivo restimulations with heterologous BCG to approximate responses elicited by the homologous strain?

4. There were 24 BCG revax subjects but PBMC from only 22 were subjected to ICS and 17 were subjected to CITE-seq. On what basis were subjects included/excluded from these analyses?

5. Line 190: missing a reference

Of these weaknesses, only #3-5 merit a response by the authors. Overall, the studies were well designed and executed. The manuscript is clear and well written. Significantly, the data are novel and provide some insights into the CD4+ response elicited by BCG revaccination or H4 boosting; such knowledge is crucial to allow us eventually to identify robust correlates of protection from TB in humans.

REVIEWER COMMENTS

Reviewer #1 (Remarks to the Author):

This manuscript looks at the proteomic and transcriptomic profile of cells taken as a separate immunogenicity study which was conducted in parallel with the POI study reported by Nemes et al NEJM 2018. A signal of protection against sustained QFT conversation was observed in the POI study for the BCG revaccination arm but not the H4/IC31 arm. The authors identify five effector memory CD4+ T-cell subpopulations associated with protection, together with a population of CD103 expressing Th17 cells.

The key limitation of this work is that the samples were taken from an immunogenicity study and there is no link or association with efficacy. The protection seen with BCG revaccination was only 45%, meaning there were also cases of sustained infection in the BCG revaccination arm. Given the heterogeneity in response seen here, it is difficult to associate any immunological signal with the protection conferred in the separate study.

A second limitation of this work is that the H4 / IC31 boost vaccination regimen did not confer significant protection against sustained infection in this study. It is not possible to distinguish between whether that was because of statistical power (i.e. the trend would have been significant with larger numbers) or whether there is no effect of this vaccine. This confounds the analysis of the H4 samples.

Neither of these two significant limitations are discussed in this manuscript.

Overall this is an interesting descriptive study presenting a detailed immunological analysis of samples from this separate clinical trial. Without an individual association with protection, the conclusions that the markers identified are correlates of protection is a bit premature.

Specific comments:

One sentence summary: should be vaccination regimen not vaccine as is BCG revaccination.

We agree with the reviewer's note here about the phrasing regarding regimen versus "vaccine" and have adjusted our wording throughout the manuscript. We also note that the one sentence summary has been removed from our revised submission to conform with Nature Communications formatting guidelines for revisions.

Line 52 – morbidity due to TB disease – not infection-

Line 59 – do the authors mean infection here? Again I suspect disease is what is meant.

We thank the reviewer for noting this inaccurate phrasing and have revised as noted in both locations.

Line 110-138 – the text describing Figure 1 does not make clear which results are statistically significant and which are not. Levels of statistical significance should be indicated in the text – for peak and also duration of response.

Also, it is important to know if there are any differences between the 3 groups in Figure 1 – at the moment only within group analysis is presented. Again statistical significance of comparison between groups is needed.

As requested, we have added sentences in this section clarifying the significance of the results for each timepoint being evaluated in Figure 1 as well as the cross-group comparisons. We have also added a Supplemental Table (Supp Table 1) to show all the requested cross-group comparisons.

Line 150-164 – it would have been interesting to see the effector memory populations stimulated with BCG /Mtb lysate after H4 immunisation to allow direct comparison between groups.

Graphs showing the BCG and *M.tb* lysate stimulation for H4:IC31 group, and the Ag85B and TB10.4 stimulation in the BCG revaccination group have been added to Figure 1b to address this point. The findings are described in the results section (Lines 163-164).

Line 350 – it is not accurate to say the H4/IC31 vaccination regimen showed potential to reduce sustained infection – this result was not statistically significant and the lack of efficacy may be real.

We agree with the reviewer that the claim here was overstated and have removed mention of H4:IC31 in this location.

Reviewer #2 (Remarks to the Author):

- Knowledge of key determinates of immunological memory underlying vaccine effectiveness is of paramount importance for the fight against infectious diseases. This work characterizes PBMC samples collected from vaccine trial participants in order to discover biomarkers and immune cell characteristics that correlate with antigen recognition, particularly CD4+ T cells

which have the capacity to effect immunological memory and associated disease resistance.

- To extensively characterize targeted subsets of PBMCs, the authors utilize single-cell analysis techniques: antibody-based multiplex protein profiling by flow cytometry; and additionally, CITE-seq which measures levels of a pre-determined panel of proteins as well as the transcriptome abundance profile of between 200 and 20,000 genes at the RNA level, for each individual cell. CITE-seq data, especially the RNA measurements are exceedingly complicated by the lack of uniform quality and completeness. A cogent transcriptomic profile, traditionally used to describe the phenotypic characteristics of a biological sample, is in single cell analysis dependent on the identification of groups of cells with relatively similar expression profiles. So while theoretically a complete protein expression profile and transcriptome are measured for each cell, in practice the data is significantly affected by random drop-outs and low dynamic range. Sophisticated statistical algorithms and best practices regarding data filtering and adjustments and calibrations, yet the application and tuning of these methods is unavoidably context-dependent and subjective. So it is important to make clear these decisions and the impact they have on the results and biological interpretation of the results.

- The authors key claim is that the post-vaccination CD4+ T cell population is qualitatively different from the subject-matched pre-vaccination. This work builds on published work by transcriptional profiling of the CD4+ cells, specifically the subset capable of expressing activation markers in response to stimulation with vaccine antigens or a pathogen (Mtb) digest.

Major issues:

- Datasets are not accessible. Code/Scripts that could be inspected or used to re-run that analysis are not provided.

The datasets have been uploaded to Figshare (DOI: 10.6084/m9.figshare.24150492). Codes have been uploaded and can be found at: https://github.com/ValentinVoillet/BCG_revax

Analysis workflow is complex enough to deserve a figure, pointing out subjective tuning steps (eg cluster consolidation decisions, relative RNA and ADT “weighting” applied in the weighted nearest neighbors, WNN, clustering).

We thank the reviewer for this helpful suggestion. This figure has been added as Supplemental Figure 3a.

- The transcriptome profile would be expected to respond more quickly to ex-vivo treatment conditions than would the protein expression profile. I don't think it is made explicitly clear that the ex-vivo stimulation has potential complication of significantly changing the transcription profile of cells. Incubation with antigen proteins or Mtb lysate for 16 hr. Genes that have a functional role in T cell activation signaling, genes that are transcriptionally regulated in response to other cells being activated or responding to treatment components. Mtb lysate may have cytotoxic or other biological activity in addition to antigens protein, potentially affecting the transcription profile. All this makes the nomenclature "antigen specific" "antigen reactive" problematic in the context of transcriptome profiling (if not protein expression). An alternative might be the term "activated" cells? And where possible the functional definition of "CD4+CD154+ subset post-stimulation"?

We thank the reviewer for their detailed consideration of our transcriptomic data. We agree that there are some potential concerns and limitations based on the stimulation conditions we opted to use and have added text acknowledging these limitations to the discussion (paragraph 2). We have also updated references to the cells we sorted for CITE-Seq as "activated" or "CD4+CD69+CD154+" or "CD8+CD69+CD137+" where appropriate.

- Line377: The kinetics and regulation differences impact not just assay sensitivity but the accuracy of gene function inferences.

This is a valid point; we have added mention of it to the discussion (paragraph 2).

- For a general audience at least, I think it would be helpful to be more systematic in the nomenclature referring to groups of cells. In vivo they might be "populations" by FACS sorting they might be (in vitro) "subsets" and in the context of proteogenomic profiling, an operational definition, (in silico) "clusters" could be used. If a novel cell type is "revealed" or "identified" by the expression profiles based clustering algorithm or heuristic, and not supported by independent characterization, indeed it would be helpful to avoid confusion of the two. Characterization of clusters could be used to make inferences about subsets. Use subset characterizations to make inferences about in vivo populations of cells.

We thank the reviewer for this helpful observation and have adopted the suggested nomenclature where feasible to reduce confusion.

- The descriptions of the hypothetical subsets represented by Clusters is hard to follow because the evidence (key genes) they refer to is buried in the supporting tables. It would be helpful to have for each cluster an executive summary listing the few key genes described for biological interpretation and the cell type assigned. The key (for gene's rna expression and/or protein primary data plotted across cohorts and time points. Expression levels and percentage cells positive can be indicated by y-scale, color, size of dot, etc. as needed. Short of biological validation, at least the verification using minimally processed data is needed in this case to depict the weight of empirical evidence for the hypothesized subset of T cells. This would lessen the burden on the technical performance of the CITE-seq data and fidelity of the data processing to support the inference from in silico cluster to potential FACS accessible subset and to pre-activation population of cells in study subjects.

We have added Figure 3b in the format suggested by the reviewer to highlight select genes that were used in annotation of the clusters. We retained the histograms showing expression of select ADTs in the different clusters in Figure 3c since we felt this type of display clearly illustrated the expression of the ADTs used to define the clusters.

- While it is reasonable to expect that the expression of proteins carefully selected to be informative with respect to known cell lineage and functions (i.e. marker proteins) would be consistent and complement the RNA expression profile, and effectively make the identity of individual cells more obvious, there is an informatic hazard that can occur when combining data from the two assays - RNA amplicons and protein ADT tags. In the organization and interpretation of clusters, the protein markers, because they are more reliably/consistently measured and easy to interpret relative to noisy RNA expression, could drive the clustering – making an “unsupervised clustering” operation insidiously supervised by the influence of domain information. I would like this to support my point that the information content of the protein and RNA data should be assessed separately, as is often the practice, carefully examine their concordance, and judiciously combine them in the analysis to understand the extent to which the clusters are driven by protein or RNA expression differences of key genes.

We thank the reviewer for raising this point. For T cells, it is known that protein quantification can provide better subset resolution than gene expression (Hao et al., 2021). In our study, WNN was able to capture key features of both modalities and showed that they were both contributing to population resolution (Figure 1, below). For instance, ADT and WNN helped us to identify additional heterogeneity within the EM CD4+ T subset with a clear separation of cluster 9 (CD103+ EM CD4+ T cells) with high ADT weights. Clusters 2, 4 and 8 were mostly defined with RNA. We have also added Supplementary Figure 3b as a visual to show the UMAPs for analysis of RNA and ADT data separately, as well as the WNN to show the combined analysis in order to make this clearer to the reader.

Figure 1: Boxplots of the modality weights (A: Gene Expression (GEX), and B: ADT) that were learned for each cell by clusters.

Minor

- The manuscript could be improved by inclusion of more data quality diagnostics such as protein/. Line 174 is very interesting but I could not find the promised details in the methods section.

We have added more detail on QC steps as well as merging of the identified clusters in the methods section.

- Terminology suggestion: Probably the “size of the cluster X” or the “proportion of cells matching the cluster X” would be more clear than the “abundance of cluster X”.

Thank you- edits made.

- Figures labels and data are not readable due in cases to poor contrast or impossibly small typeface.

We sympathize with the reviewer here and have tried to enlarge typeface and re-arrange some figures where possible to increase the size of panels that contain small text (e.g., Figure 3d)

Proportions are sometimes reported as % and sometimes as fractions so where the y axis values labels or data reach only to 1.

Thank you for pointing this out. Supplemental Tables and Figures have been corrected and made uniform.

- MAST differential expression outputs reported in supp tables do not always have p-value and adjusted p-value. They also propagate the error where the default outputs from the software label a fractional quantity as a percent ‘pct’. One can change the description in the table description to provide the correction.

Thank you for pointing this out. The fractional quantities have been corrected accordingly.

- I believe the proper nomenclature for gene expression, as measured by RNA abundance, is the uppercase non-italicized. Referring to the GENE and not a transcript. The manuscript has lower-case italics.

We thank the reviewer for flagging this. We have adjusted the formatting to follow the guidance of the most authoritative source we could find on the topic, the HUGO Gene Nomenclature Committee. They advise the use of uppercase and italicized for genes and their mRNA transcripts, with non-italicized uppercase used only for protein products.

- T cell receptor sequencing v(d)j analysis was mentioned but not analyzed or data provided.

The TCR sequence data has been included in the data repository for this paper and analysis has been included in the manuscript.

- CD8+ subset CITE-seq analysis is not presented or data provided.

Analysis of CD8+ T cell CITE-Seq data is now included as Supplemental Figure 7 and Supplemental Tables 8 and 9.

- L627 how many cells were dropped due to low ADT counts? What was the range exactly?

This was an omission on our part. We have now included a statement that only 57 cells were excluded from analysis based on our QC criterion for ADT expression.

- L634 Please include analysis code or post on github.

The analysis code can now be found at: https://github.com/ValentinVoillet/BCG_revax

- L656 Please provide more details about criteria for deciding the number of clusters were optimal for the dataset.

Additional detail has been added to the methods section describing this.

- L687 fold change, log fold change, base 10 or base2, pFDR, FDR p-value, FDRq value – these terms are inconsistently used in the manuscript – It is hard to determine if these are typos. 5% or 0.05. Also the use of FDR at different significance level cutoffs, for 20 tests. Theoretically FDR for the top most significant result (of all tests) is just the p-value?

We agree that it is best to be consistent with these terms. We have revised all references to “FDR” to use the term “FDR q-value”, which is the Benjamini and Hochberg FDR value.

Throughout the manuscript we have used $FDR < 0.05$ as the cut-off for significant results that we report in the manuscript. For the serum cytokine analysis we also included reporting of several additional cytokines that were $FDR < 0.20$; the Supplementary Figure and Table as well as the Results text list these cut-offs. Though any cut-off is arbitrary, the benefit of using an FDR cutoff is that the interpretation is clear (e.g., we expect that 0.2 proportion of the data is a false-discovery when we accept $FDR\ q\text{-value} < 0.2$ is a discovery). Since this is a small study and the quantification of low concentration cytokines is an exploratory objective, we thought it was acceptable to use $FDR\ q\text{-value} < 0.2$ as a “hypothesis generating” significance criterion. We have also added this explanation to the Methods section.

We note that in the analysis to identify genes and ADTs that defined individual clusters; we presented p-values and FWER-adjusted p-values, as is customary for this kind of analysis, since the unit of analysis is single cells. We have explicitly noted in the table that these adjusted p-values are FWER-adjusted.

- L1142 Color scale legends for heatmaps are sometimes missing numeric values.

Thank you for pointing this out, the scales and legends have been added to heatmaps.

- Supp Tables have “Excel mutated” gene names appearing – apparently upstream corruption because gene descriptions are missing for those non-gene gene names.

We apologize for the inadvertent data munging in the tables. These issues have been corrected.

- Supp tables are sometimes missing raw p-value or adjusted p-value

Raw p-values or adjusted p-values have been added in as appropriate where previously omitted in the supplementary tables.

- L1064 What was the rate of detection of each of the effector genes?

We interpreted this question to be asking about expression of each of the functional markers for the antigen-reactive cells expressing 1 of 7 functional markers we selected for analysis. The rate of detection for each of these proteins in the ICS assay is listed below as a percentage of all cells selected for analysis (across vaccine groups, timepoints and participants), however we believe the addition of this information to the manuscript without further analysis or contextualization on its own is of limited value. The clustering analysis in Figure 2 shows expression of these proteins individually by cluster and identifies those that are modulated by vaccination.

Sorted by order of most to least expressed, for the CD4+ T cells expressing at least 1 of 7 functions:

- TNF- α is expressed in 54% of the CD4+ T cells.
- CD154 is expressed in 43% of the CD4+ T cells.
- IL-2 is expressed in 26% of the CD4+ T cells.
- IL-17a is expressed in 24% of the CD4+ T cells.
- IFN- γ is expressed in 16% of the CD4+ T cells.
- IL-22 is expressed in 2% of the CD4+ T cells.
- IL-4/13 is expressed in 0.5% of the CD4+ T cells.

- Supp figures L42 Typo, should be CD8+CD69+CD137+ ?

Thank you, this typographical error has been corrected.

Reviewer #3 (Remarks to the Author):

In this submission by Dintwe, et al., the authors extend their initial report of the HVTN 602/Aeras A-042 trial (EClinical Medicine, 2020) to better characterize the CD4+ T cell response in South African adolescents to BCG revaccination or boosting with H4:IC31. The experimental approach is sound, using both multiparameter flow cytometry and CITE-Seq, as well as appropriate data analytics. Unsupervised cluster analysis of flow cytometric data revealed several CD4+ TEM populations that emerged especially after BCG revaccinations, two of which were polyfunctional, including one with Th1* features. Both vaccine regimens increased ag-specific TEM cells and CITE-Seq revealed several distinct TEM populations, including some with migratory potential and a small cluster of Th17+ cells that expressed high levels of CD103, suggesting these cells may represent TRM cells, shown in NHP models to correlate with protection in the lung. The manuscript has several strengths:

1. It carefully and comprehensively reveals the heterogenous CD4+ T cell response to these vaccines. While it is impossible to correlate specific populations with protection in this study, this work does provide valuable data that can be integrated with data from other TB vaccine trials.
2. Kudos to the authors for not just focusing on the cells themselves, but also measuring cyto/chemokines in supes, which provided confirmation of gene expression analyses (e.g. IFN- γ).
3. Care must be taken when interpreting UMAP plots and WNN analysis, and the authors take care not over-interpret clusters consisting of few cells.

There are a few issues that limit the significance of the current manuscript.

1. As with most vaccine trials, the samples consisted of only cryopreserved PBMC; no airway cells (i.e. BAL) were collected. The authors do clearly acknowledge this limitation in the discussion.
2. There was no responses in CD8 T cells detected so these were excluded from further analysis. It is likely that the vaccine regimens did indeed induce a CD8 response as shown by other studies of TB vaccines in humans. As acknowledged in the discussion, the failure to detect

such a response here was likely due to assays conditions that favor CD4+, rather than CD8+, responses, but still represents a missed opportunity.

We thank the reviewer for this comment. We have included the CD8+ T-cell data in Supplementary Figure 7 for the CITE-Seq analysis.

3. Subjects in the revaccination group received BCG SSI. But BCG Pasteur was used for in vitro restimulation here (as well as in the EClin Med paper). Why was BCG Danish not used for restimulation? Is it reasonable to expect ex vivo restimulations with heterologous BCG to approximate responses elicited by the homologous strain?

At the time of commencing this study, there was a worldwide shortage of the BCG Danish strain produced by Statens Serum Institut (SSI), and even South African clinics were unable to procure the vaccine for vaccination of infants. This point has been addressed in the discussion.

4. There were 24 BCG revax subjects but PBMC from only 22 were subjected to ICS and 17 were subjected to CITE-seq. On what basis were subjects included/excluded from these analyses?

For some trial participants, there were limited PBMC vials available due to low-volume blood draws at the clinic. The ICS assays addressed a primary endpoint of the HVTN 602 trial and therefore PBMC were prioritized for this assay. After the ICS assays were performed, we only had enough vials of PBMC at the correct timepoints to perform CITE-Seq assays on 17 participants. This information has been added to the Methods.

5. Line 190: missing a reference

Thank you for flagging this, the reference is now inserted.

Of these weaknesses, only #3-5 merit a response by the authors. Overall, the studies were well designed and executed. The manuscript is clear and well written. Significantly, the data are novel and provide some insights into the CD4+ response elicited by BCG revaccination or H4 boosting; such knowledge is crucial to allow us eventually to identify robust correlates of protection from TB in humans.

REVIEWERS' COMMENTS

Reviewer #1 (Remarks to the Author):

The authors have addressed many of the reviewers comments but have not yet addressed the two key limitations of this study in either the introduction or the discussion:

The key limitation of this work is that the samples were taken from an immunogenicity study and there is no link or association with efficacy. The protection seen with BCG revaccination was

only 45%, meaning there were also cases of sustained infection in the BCG revaccination arm.

Given the heterogeneity in response seen here, it is difficult to associate any immunological signal with the protection conferred in the separate study.

A second limitation of this work is that the H4 / IC31 boost vaccination regimen did not confer

significant protection against sustained infection in this study. It is not possible to distinguish between whether that was because of statistical power (i.e. the trend would have been

significant with larger numbers) or whether there is no effect of this vaccine. This confounds the analysis of the H4 samples.

Neither of these two significant limitations are discussed in this manuscript.

Reviewer #2 (Remarks to the Author):

With their thoughtful responses and manuscript revisions the authors have fully addressed all of my concerns.

Reviewer #3 (Remarks to the Author):

This submission extends the initial report from this group of the HVTN 602/Aeras A-042 trial to better characterize the CD4+ T cell response in South African adolescents to BCG

revaccination or boosting with H4:IC31. The experimental approach is sound, using both multiparameter flow cytometry and CITE-Seq, as well as appropriate data analytics. Unsupervised cluster analysis of flow cytometric data revealed several CD4+ TEM populations that emerged especially after BCG revaccinations, two of which were polyfunctional, including one with Th1* features. Both vaccine regimens increased ag-specific TEM cells and CITE-Seq revealed several distinct TEM populations, including some with migratory potential and a small cluster of Th17+ cells that expressed high levels of CD103, suggesting these cells may represent TRM cells, shown in NHP models to correlate with protection in the lung. The manuscript has several strengths:

1. It carefully and comprehensively reveals the heterogenous CD4+ T cell response to these vaccines. While it is impossible to correlate specific populations with protection in this study, this work does provide valuable data that can be integrated with data from other TB vaccine trials.
2. The authors did not just focus on the cell types, but also measured cyto/chemokines in supes, supplementing the gene expression analyses (e.g. IFN-g).
3. The authors took care when interpreting UMAP plots and WNN analyses and did not over-interpret clusters of a few cells.

A few issues limit the significance of this work.

1. As with most vaccine trials, the samples consisted of only cryopreserved PBMC; no airway cells (i.e. BAL) were collected.
2. The analysis of CD8+ T cells was limited to the CITE-Seq data in Supp. Fig. 7. CD8+ T cell responses are likely important for TB vaccine-elicited protection but, as acknowledged in the discussion, the assay conditions favor CD4+, rather than CD8+, responses.
3. Subjects in the revaccination group received BCG SSI, but BCG Pasteur was used for in vitro restimulation. The reason for using heterologous BCG (Shortage of BCG Danish) is now explained.

This study was well designed and executed, and the manuscript is well written. Significantly, the data are novel and provide some insights into the CD4+ response elicited by BCG revaccination or H4 boosting. Such knowledge is crucial to identify robust correlates of protection from TB in humans.

REVIEWERS' COMMENTS:

Reviewer #1 (Remarks to the Author):

The authors have addressed many of the reviewers comments but have not yet addressed the two key limitations of this study in either the introduction or the discussion:

The key limitation of this work is that the samples were taken from an immunogenicity study and there is no link or association with efficacy. The protection seen with BCG revaccination was only 45%, meaning there were also cases of sustained infection in the BCG revaccination arm. Given the heterogeneity in response seen here, it is difficult to associate any immunological signal with the protection conferred in the separate study.

A second limitation of this work is that the H4 / IC31 boost vaccination regimen did not confer significant protection against sustained infection in this study. It is not possible to distinguish between whether that was because of statistical power (i.e. the trend would have been significant with larger numbers) or whether there is no effect of this vaccine. This confounds the analysis of the H4 samples.

Neither of these two significant limitations are discussed in this manuscript.

Author response:

The reviewer's comments are noted. We have updated text on lines 527-531 of the Discussion to emphasize that our results were obtained from the phase 1b HVTN 602 trial and were therefore limited to an immunogenicity assessment, but that our work has identified populations that can be analyzed in the planned case-control study of the C-040-404 trial. We mention in the introduction that "H4:IC31 vaccination showed a similar trend that did not reach significance (efficacy of 30.5%, $P=0.29$)" but believe it is beyond the scope of the current manuscript to speculate on the reasons significance was not reached in the H4:IC31 arm of the larger trial.